# Privacy Preserving Reinforcement Learning for Population Processes

**Samuel Yang-Zhao**                                     *samuel.yang-zhao@anu.edu.au*
*Australian National University*

**Kee Siong Ng**                                         *keesiong.ng@anu.edu.au*
*Australian National University*

**Reviewed on OpenReview:** *https://openreview.net/forum?id=zZFb1aDUeE*

## Abstract

We consider the problem of privacy protection in Reinforcement Learning (RL) algorithms that operate over population processes, a practical but understudied setting that includes, for example, the control of epidemics in large populations of dynamically interacting individuals. In this setting, the RL algorithm interacts with the population over $T$ time steps by receiving population-level statistics as state and performing actions which can affect the entire population at each time step. An individual's data can be collected across multiple interactions and their privacy must be protected at all times. We clarify the Bayesian semantics of Differential Privacy (DP) in the presence of correlated data in population processes through a Pufferfish Privacy analysis. We then give a meta algorithm that can take any RL algorithm as input and make it differentially private. This is achieved by taking an approach that uses DP mechanisms to privatize the state and reward signal at each time step before the RL algorithm receives them as input. Our main theoretical result shows that the value-function approximation error when applying standard RL algorithms directly to the privatized states shrinks quickly as the population size and privacy budget increase. This highlights that reasonable privacy-utility trade-offs are possible for differentially private RL algorithms in population processes. Our theoretical findings are validated by experiments performed on a simulated epidemic control problem over large population sizes.

## 1 Introduction

The increasing adoption of Reinforcement Learning (RL) algorithms in many practical applications such as digital marketing, finance, and public health (Mao et al., 2020; Wang & Yu, 2021; Charpentier et al., 2020) have led to new, challenging privacy considerations for the research community. This is a particularly important issue in domains like healthcare where highly sensitive personal information is routinely collected and the use of such data in training RL algorithms must be handled carefully, in light of successful privacy attacks on RL algorithms (Pan et al., 2019). Privacy Preserving Reinforcement Learning is an active research area looking to address these concerns, mostly via the now widely accepted concept of Differential Privacy (DP) (Dwork et al., 2006), which confers formal 'plausible deniability' guarantees for users whose data are used in training RL algorithms, thus offering them privacy protection.

In this paper, we consider the setting where an RL agent interacts with a population of individuals and investigate how each individual's differential privacy can be protected. We assume interactions between individuals are not visible to the RL agent but the agent receives population-level statistics at every time step and can perform actions that affect the entire population. This class of environments, which we call *population processes*, models settings such as the control of epidemic spread by government interventions (Kompella et al., 2020). As an individual's data is collected across multiple interactions, our goal is to ensure that an individual's data contributions over all interactions are differentially private. To the best of our

knowledge, existing approaches to privacy-preserving reinforcement learning, which we survey next in § 1.1, do not cater to this natural and important problem setting and are unable to exploit the structure of the problem.

## 1.1 Related Work

The earliest works to consider differential privacy in a reinforcement learning context were focused on the bandit or contextual bandit settings (Guha Thakurta & Smith, 2013; Mishra & Thakurta, 2015; Tossou & Dimitrakakis, 2016; 2017; Shariff & Sheffet, 2018; Sajed & Sheffet, 2019; Zheng et al., 2020; Dubey & Pentland, 2020; Ren et al., 2020; Chowdhury & Zhou, 2022b; Azize & Basu, 2022). A key negative result is in Shariff & Sheffet (2018), who show that sublinear regret is not possible under differential privacy in contextual bandits, a result which also holds in the reinforcement learning setting.

Differentially private reinforcement learning beyond the bandit setting has been primarily considered in a personalization context. In Balle et al. (2016), the authors study policy evaluation under the setting where the RL algorithm receives a set of trajectories, and a neighbouring dataset is one in which a single trajectory differs. In a regret minimization context, there is a body of work on designing RL algorithms to satisfy either joint differential privacy (Vietri et al., 2020; Luyo et al., 2021; Chowdhury & Zhou, 2022a; Ngo et al., 2022; Zhou, 2022; Qiao & Wang, 2023) or local differential privacy (Garcelon et al., 2020; Luyo et al., 2021; Chowdhury & Zhou, 2022a; Liao et al., 2023). In all of these works, the RL algorithm is framed as interacting with users in trajectories or episodes, with each trajectory representing multiple interactions with a *single* user. A neighbouring dataset is then defined with respect to a neighbouring trajectory. Such DP-RL approaches cannot be easily adapted for privacy protection in population processes, where (i) each interaction is with an entire population (rather than a single individual); (ii) a specific individual is typically not sampled in consecutive time steps so there is not a corresponding notion of a trajectory; and (iii) an individual's data could actually be present across all time steps, a low-probability event that we nevertheless have to handle.

In Wang & Hegde (2019), the authors analyse the performance of deep Q-learning (Mnih et al., 2015) under differential privacy guarantees specified with respect to neighbouring reward functions. This notion of privacy makes natural sense when the reward function is viewed as an individual's private preferences but is also inapplicable to our setting as it does not consider the privacy of the state.

In relation to our application domain and experimental results, the control of epidemics is a topical subject given the prevalence of COVID-19 in recent years and many different practical approaches have been developed (Arango & Pelov, 2020; Charpentier et al., 2020; Colas et al., 2020; Berestizshevsky et al., 2021; Kompella et al., 2020; Chen et al., 2022). Whilst the preservation of individual privacy has been considered in the context of public release of population-level epidemic statistics (Dyda et al., 2021; Li et al., 2023), we are not aware of other work that achieves individual privacy preservation in the modelling and control of epidemics using reinforcement learning.

## 1.2 Contributions

The main questions we address in this paper are the following:

1. For an RL algorithm interacting with a population of individuals, what is the right notion of individual privacy we should care about, and can that individual privacy be protected using differential privacy?

2. Assuming the answer to the first question is positive, are good privacy-utility trade-offs possible for differentially private RL algorithms in population processes?

We answer the first question in two parts. We first show through a concrete example (see Example 4) that highly correlated data is possible in population processes and that some sensitive information that one may naturally wish to protect, like a person's infection status during an epidemic, cannot always be protected. We then give, using the general Pufferfish Privacy framework (Kifer & Machanavajjhala, 2014), a precise semantic definition of the secrets around individual participation in samples that can actually be protected and show its equivalence to $k$-fold adaptive composition of DP mechanisms (see Lemma 1). Building on that,

we then describe a family of DP-RL algorithms for population processes and show how differential privacy and correlated data are somewhat orthogonal issues in our set up.

On the second question, we note our DP-RL solution is a meta algorithm that takes any RL algorithm (whether online/offline or value-based/policy-based) as a black box and make it differentially private. Whilst such a modular solution is desirable for its simplicity and generality, the trade-off is a more difficult control problem because the underlying environment becomes partially observable to the RL agent. Standard methods for dealing with partial observability typically require expensive state-estimation techniques or sampling based approximations (Monahan, 1982; Shani et al., 2013; Kurniawati, 2022). Instead of using these methods, we analyze the performance of standard RL algorithms as the sampled population size $N$ and privacy budget $\epsilon$ increases. Under some assumptions, we obtain the following bound on the approximation error under privacy

$$\left\|Q^* - \tilde{Q}^*\right\|_\infty \leq \mathcal{O}\left(\sqrt{K}\exp\left(-\frac{\epsilon}{2\sqrt{2K}}\right) + K\exp\left(-\sqrt{N}\epsilon\right) + \frac{K^{\frac{3}{2}}}{\sqrt{N}}\right), \tag{1}$$

where $Q^*$ is the optimal value function for a given (arbitrary) population process $M$, $\tilde{Q}^*$ is the optimal value function for the privatized form of $M$, and $K$ is the dimension of the state space. (The precise statement is Theorem 3 in § 4.2.) Note that the RHS of (1) goes to 0 as $N$ and $\epsilon$ increases, and that such a result is not possible in the personalization setting (Shariff & Sheffet, 2018). Whilst this result does not imply a finite-time sample complexity guarantee, we validate empirically on an epidemic control problem simulated over large graphs that our DP-RL algorithm behaves well as the population size and privacy budget increases, as suggested by (1). Our results demonstrate that reasonable privacy-utility trade-offs are certainly possible for differentially private RL algorithms in population processes.

## 2 Preliminaries

### 2.1 Reinforcement Learning and Markov Decision Processes

We consider a time-homogeneous Markov Decision Process (MDP) $M = (\mathcal{S}, \mathcal{A}, P, r, \gamma)$ with state space $\mathcal{S}$, action space $\mathcal{A}$, transition function $P : \mathcal{S} \times \mathcal{A} \to \mathcal{D}(\mathcal{S})$, reward function $r : \mathcal{S} \times \mathcal{A} \to [0, r_{\max}]$, discount factor $\gamma \in [0, 1)$. The notation $\mathcal{D}(\mathcal{S})$ defines the set of distributions over $\mathcal{S}$. The rewards are assumed to be bounded between 0 and $r_{\max} \in \mathbb{R}$. A stationary policy is a function $\pi : \mathcal{S} \to \mathcal{D}(\mathcal{A})$ specifying a distribution over actions based on a given state, i.e. $a_t \sim \pi(\cdot \,|\, s_t)$. A stationary deterministic policy assigns probability 1 to a single action in a given state. With a slight abuse of notation, we will define a stationary deterministic policy to have the signature $\pi : \mathcal{S} \to \mathcal{A}$. Let $\Pi$ denote the set of all stationary policies. The action-value function (Q-value) of a policy $\pi$ is the expected cumulative discounted reward $Q^\pi(s, a) = r(s, a) + \mathbb{E}_{P,\pi}\left[\sum_{t=1}^\infty \gamma^t r(s_t, a_t)\right]$, where the expectation is taken with respect to the transition function $P$ and policy $\pi$ at each time step. The value function is defined as $V^\pi(s) = \mathbb{E}_{a\sim\pi(\cdot\,|\,s)}[Q^\pi(s, a)]$. The Q-value satisfies the Bellman equation given by $Q^\pi(s, a) = r(s, a) + \gamma\mathbb{E}_P\left[V^\pi(s')\right]$.

When considering the optimal policy, define $V^*(s) = \sup_{\pi\in\Pi} V^\pi(s)$ and $Q^*(s, a) = \sup_{\pi\in\Pi} Q^\pi(s, a)$. The optimal action-value function satisfies the bellman optimality equation given by $Q^*(s, a) = r(s, a) + \gamma\mathbb{E}_P\left[\max_{a'} Q^*(s', a')\right]$. There exists an optimal stationary deterministic policy $\pi^* \in \Pi$ such that $V^*(s) = V^{\pi^*}(s)$. The greedy policy $\pi^*(s) = \arg\max_a Q^*(s, a)$ is in fact an optimal policy.

We primarily consider the case when $\mathcal{S}$ is finite. In this case, it is helpful to view our functions as vectors and matrices. We use $P$ to refer to a matrix of size $(|\mathcal{S}| \times |\mathcal{A}|) \times |\mathcal{S}|$ where $P_{sa}^{s'}$ is equal to $P(s' \,|\, s, a)$ and $P_{sa}$ is a length $|\mathcal{S}|$ vector denoting $P(\cdot \,|\, s, a)$. Similarly, we can view $V^\pi$ as a vector of length $|\mathcal{S}|$ and $Q^\pi$ and $r$ as vectors of length $|\mathcal{S}| \times |\mathcal{A}|$. The Bellman equation can now be expressed as

$$Q^\pi = r + \gamma P V^\pi.$$

### 2.2 Stochastic Population Processes

A stochastic population process models a stochastic system evolving over a collection of individuals and allows for heterogeneous interactions between them. Consider a population of $N^*$ individuals indexed by

the set $[N^*] = \{1, \ldots, N^*\}$. Individual $i$'s status at time $t$ is given by the random variable $X_{t,i} \in [K]$, which takes one of $K$ values, and $X_t = (X_{t,1}, \ldots, X_{t,N^*})$ denotes the random vector of all individuals' status. A graph at time $t$ is denoted by $G_t = (\mathcal{V}, \mathcal{E}_t)$ where the nodes $\mathcal{V}$ represent the individuals and $\mathcal{E}_t$ represent the interactions between individuals at time $t$. We assume that the total number of individuals is fixed but the edges evolve over time. To denote sequences, let $Y_{1:t} = (Y_1, \ldots, Y_t)$ and $Y_{<t} = (Y_1, \ldots, Y_{t-1})$. The graph at time $t$ evolves according to a distribution $G_t \sim P(\cdot \,|\, G_{<t})$. An individual's status depends only upon the previous graph and the previous status of its neighbours, thus satisfying the Markov property, and can be expressed as $P(X_t \,|\, G_{<t}, X_{<t}) = P(X_t \,|\, G_{t-1}, X_{t-1})$. We will assume that individuals' initial status are drawn independently from a distribution $P(X_0 \,|\, G_{<0}, X_{<0}) = P(X_0)$ and are not dependent on any interaction graph.

## 2.3 Differential Privacy

Differential privacy is now a commonly accepted definition of privacy with good guarantees even under adversarial settings (Dwork et al., 2006). The differential privacy definition allows different notions of privacy by defining the concept of a neighbouring dataset appropriately. The following definition is a common choice (Dimitrakakis et al., 2017):

**Definition 1** (Differential Privacy)**.** A randomized mechanism $\mathcal{M} : \mathcal{X}^n \to \mathcal{U}$ is $(\epsilon, \delta)$-differentially private if for any $D \in \mathcal{X}^n$ and for any measurable subset $\Omega \subseteq \mathcal{U}$

$$P(\mathcal{M}(D) \in \Omega) \leq e^\epsilon P(\mathcal{M}(D') \in \Omega) + \delta,$$

for all $D'$ in the Hamming-1 neighbourhood of $D$. That is, $D'$ may differ in at most one entry from $D$: there exists at most one $i \in [n]$ such that $D_i \neq D'_i$.

A standard approach to privatising a query over an input dataset is to design a mechanism $\mathcal{M}$ that samples noise from a carefully scaled distribution and add it to the true output of the query. To scale the noise level appropriately, the sensitivity of a query is an important parameter.

**Definition 2** ($\ell_1$ sensitivity)**.** Let $f$ be a function $f : \mathcal{X}^n \to \mathcal{U}$. Let $d : \mathcal{X}^n \times \mathcal{X}^n \to \{0, 1\}$ be a function indicating whether two inputs are neighbours. The sensitivity of $f$ is defined as $\Delta_f = \sup_{x,x' \in \mathcal{X}^n : d(x,x')=1} \|f(x) - f(x')\|_1$.

Differential privacy has several well known properties, such as composing adaptively and being preserved under post-processing, that we will make use of.

## 2.4 Pufferfish Privacy.

Pufferfish privacy was introduced in Kifer & Machanavajjhala (2014) and proposes a generalization of differential privacy from a Bayesian perspective. In the Pufferfish framework, privacy requirements are instantiated through three components: (i) $\mathbb{S}$, the set of secrets representing functions of the data that we wish to protect; (ii) $\mathbb{Q} \subseteq \mathbb{S} \times \mathbb{S}$, a set of secret pairs that need to be indistinguishable to an adversary; and (iii) $\Theta$, a class of distributions that can plausibly generate the data. Typically $\Theta$ is viewed as the beliefs that an adversary holds over how the data was generated. Pufferfish privacy is defined as follows:

**Definition 3** (Pufferfish Privacy)**.** Let $(\mathbb{S}, \mathbb{Q}, \Theta)$ denote the set of secrets, secret pairs, and data generating distributions and let $D$ be a random variable representing the dataset. A privacy mechanism $\mathcal{M}$ is said to be $(\epsilon, \delta)$-Pufferfish private with respect to $(\mathbb{S}, \mathbb{Q}, \Theta)$ if for all $\theta \in \Theta$, $D \sim \theta$, for all $(s_i, s_j) \in \mathbb{Q}$, and for all $w \in Range(\mathcal{M})$, we have

$$e^{-\epsilon} \leq \frac{P(\mathcal{M}(D) = \omega \,|\, s_i, \theta) - \delta}{P(\mathcal{M}(D) = \omega \,|\, s_j, \theta)} \leq e^\epsilon, \tag{2}$$

where $s_i, s_j$ are such that $P(s_i \,|\, \theta) \neq 0, P(s_j \,|\, \theta) \neq 0$.

For $\delta = 0$, applying Bayes Theorem to Equation 2 shows that Pufferfish privacy can be interpreted as bounding the odds ratio of $s_i$ to $s_j$; an attacker's belief in $s_i$ being true over $s_j$ can only increase by a factor

of $e^\epsilon$ after seeing the mechanism's output. The main advantages of Pufferfish privacy are that it provides a formal way to explicitly codify what privacy means and what impact the data generating process has. This has been used to clarify, among other things, the semantics of differential privacy protection in the presence of possibly correlated data in Kifer & Machanavajjhala (2014), a topic that motivated multiple attacks and possible solutions (Kifer & Machanavajjhala, 2011; Liu et al., 2016; Srivatsa & Hicks, 2012; Zhao et al., 2017; Yang et al., 2015; Almadhoun et al., 2020). We will similarly use the Pufferfish framework to make the semantics of our privacy guarantees in population processes, which can have correlated data, explicit.

## 3 Problem Setting

### 3.1 Model

We now describe how the problem of controlling population processes is modelled as an MDP and also the underlying data-generation process.

The environment $\mathscr{E}$ is modelled as a stochastic population process evolving over $N^*$ individuals. The RL agent is denoted by $\mathscr{A}$. We assume there is a trusted data curator $\mathscr{D}$ that collects the data from the environment. At each time step, $N$ individuals are randomly sampled (not necessarily uniformly) and their potentially sensitive data is collected by the data curator. We consider the case where the interactions between individuals are unknown, i.e. $\mathscr{D}$ has no access to the graph sequence $G_{1:T}$. This models many problem domains such as the control of epidemics where the underlying interactions between people in a population are not visible and decisions can only be made based upon population statistics. We consider the case of histogram queries. The agent $\mathscr{A}$ picks actions at each time step depending upon the received state and also computes its reward $r_t = r(s_t, a_{t-1})$ as a function of the current state and previous action. In summary, the environment, agent, and data curator interact in the following manner, given an initial graph $G_0$ and status $X_0$ generated from some distribution.

For time $t = 1, \ldots, T$:

1. $\mathscr{E}$ generates the status for all individuals $X_t \sim P(\cdot \,|\, G_{t-1}, X_{t-1})$, with $X_t = (X_{t,1}, \ldots, X_{t,N^*})$.

2. $\mathscr{D}$ samples a subset $L_t \subseteq [N^*]$ of size $N$ and produces the dataset $D_t = (X_{t,i})_{i \in L_t}$.

3. $\mathscr{D}$ answers histogram query $s_t = q(D_t) = \frac{1}{N} \sum_{i \in L_t} (\mathbb{I}(X_{t,i} = \alpha))_{\alpha \in [K]}$ and computes reward $r_t = r(s_t, a_{t-1})$.

4. $\mathscr{A}$ receives state $s_t$ and reward $r_t$ and forms the transition sample $(s_{t-1}, a_{t-1}, s_t, r_t)$ to learn from.

5. $\mathscr{A}$ picks the next action $a_t \sim \pi_t(\cdot \,|\, s_t)$.

6. The graph $G_t \sim P(\cdot \,|\, G_{t-1}, a_t)$ is sampled depending on the last graph and action selected.

The state space $\mathcal{S}$ thus consists of any vector $z \in [0,1]^K$ that satisfies $\sum_{i \in [K]} z_i = 1$ and $z_i = c_i/N$ for some $c_i \in \{0, 1, \ldots, N\}$. The action space $\mathcal{A}$ is application-dependent. In general, we consider the setting where a selected action $a_t \in \mathcal{A}$ modifies the edges in the graph. For example an action could correspond to cutting all edges for a subset of nodes, emulating the effect of 'quarantining' individuals in an epidemic control context. Thus the graph at time $t$ depends upon the action chosen and is distributed according to $P(G_t \,|\, G_{t-1}, a_t)$. These underlying transitions on individuals' states and the graph structure will induce a transition function $P : \mathcal{S} \times \mathcal{A} \to \mathcal{D}(\mathcal{S})$ over the state space that implicitly captures the impact of the agent's actions.

An individual's data is exposed via the state histogram query $s_t = q(D_t)$. The state query has sensitivity $\Delta_q = 2/N$ as using individual $j$'s data instead of individual $i$'s data can change the counts in at most two bins of the histogram. Additionally an individual could be sampled multiple times by $\mathscr{D}$ over $T$ interactions so we need to ensure that their combined data over $T$ steps are treated in a differentially private way.

**Example 1** (Epidemic Control). Throughout this paper we consider the Epidemic Control problem as a concrete instantiation of our problem setting. One particular example is the Susceptible-Exposed-Infected-Recovered-Susceptible (SEIRS) process on contact networks (Pastor-Satorras et al., 2015; Nowzari et al.,

2016; Newman, 2018). An SEIRS process on contact networks is parametrized by a graph $G = (\mathcal{V}, \mathcal{E})$, representing interactions, and four transition rates $\beta, \sigma, \gamma, \rho > 0$. At any point in time, each individual is in one of four states: Susceptible (S), Exposed (E), Infected (I), or Recovered (R). If individual $i$ is Susceptible at time $t$ and has interacted with $d_{t,i}$ individuals who are Infected, then individual $i$ becomes Exposed with probability $1 - (1 - \beta)^{d_{t,i}}$. Once Exposed, an individual becomes Infected after Geometric($\sigma$) amount of time. Similarly, an Infected individual becomes Recovered with Geometric($\gamma$) time and a Recovered individual becomes Susceptible again with Geometric($\rho$) time. At each time $t$, the state is a histogram representing the proportion of individuals that are of each status. Instead of allowing all interactions, the agent's actions allow for a subset of nodes to be quarantined for one time step. This has the effect of modifying the graph's edges such that quarantined individuals have no edges for a single time step. A typical reward function provides a cost proportional to the number of individuals quarantined and the number of infected individuals.

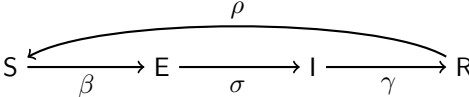

Figure 1: Visualisation of the parameters that govern the transitions between states for individuals in the SEIRS process over contact networks.

**Example 2** (Countering Misinformation). The spread of misinformation in online social media is one of the key threats to society. There is good literature on different (mis)information percolation and diffusion models (Del Vicario et al., 2016; Van Der Linden, 2022) that take factors like homogeneity and polarisation into account, as well as containment strategies like running counter-information campaigns to debunk or 'prebunk' misinformation, possibly through targeting of influencers in social networks (Budak et al., 2011; Nguyen et al., 2012; Acemoglu et al., 2023; Ariely, 2023). Designing reinforcement learning algorithms that can detect and control spread of misinformation in a differentially private manner is another example of our general problem setting.

**Example 3** (Malware Detection and Control). Malware propagation models on large-scale networks (Yu et al., 2014) and smartphones (Peng et al., 2013) have long been a subject of interest in cyber security, with more recent work focussing on malware propagation in internet-of-things (Li et al., 2020; Yu et al., 2022; Muthukrishnan et al., 2021). Designing reinforcement learning algorithms to detect and control the spread of malware, especially unknown malwares whose signatures can only be discerned from collecting data on potentially sensitive device-usage behaviour, is also an example of our general problem setting.

### 3.2 Formalizing Privacy in the Presence of Correlated Data

In this section we clarify the precise privacy protections we will provide under differential privacy in our problem setting. Protecting privacy in population processes presents some unique issues that are not typically encountered in the standard application of differential privacy. In particular, correlations between different individuals' data can easily arise due to the fact that individuals interact with each other at each time step and also over multiple time steps. Whilst differential privacy's guarantees are a mathematical statement that hold regardless of the process that generated the data, the semantics of these guarantees are often misinterpreted when it is applied. This leads to misalignment between what one hopes to keep private and what is actually kept private. We illustrate this with an example.

**Example 4.** Consider a highly contagious but long recovery flu spreading in a tight-knit community of $N^*$ individuals who live in the same location. The dataset is $D_t = (X_{t,i})_{i \in L_t}$ where $L_t \subseteq [N^*]$ denotes a subset of individuals sampled for their flu status $X_{t,i} \in \{0, 1\}$. As the individuals live in the same location, their interaction graph is a fully connected graph. The goal is to release the number of infected individuals $q_t = \sum_{i \in L_t} X_{t,i}$ at time $t$ whilst still preventing an adversary from detecting whether a particular individual, say Alice, has the flu at that point in time. If the underlying data generating process is ignored, the statistic $q_t$ has sensitivity 1 and $\tilde{q}_t = q_t + Lap(1/\epsilon)$, where $Lap(1/\epsilon)$ is the noise generated from the Laplace mechanism, is an $\epsilon$-differentially private statistic. However, given the flu's characteristics and the fully connected interaction graph, we can say with high probability either all individuals or no individuals are infected at any time $t$.

Thus, an adversary could guess from the value of $\tilde{q}_t$ which of these two scenarios is the case, thereby also recovering Alice's flu status with high probability.

It would be natural to hope that an individual's flu status could be protected under differential privacy but Example 4 illustrates that this may not be possible if the data is correlated and the adversary has additional information about the problem. Solutions like Group Differential Privacy (Dwork & Roth, 2014), which adds noise proportional to the largest connected component in a graph, and Dependent Differential Privacy (Liu et al., 2016), which adds noise proportional to the amount of correlation in the data, exist but the amount of additional noise that is required typically destroys utility. So what can differential privacy protect in population processes in general and in special scenarios like Example 4? The answer, as we shall see, is that differential privacy confers plausible deniability on an individual's participation or presence in a dataset sampled from the underlying population, but not the individual's actual status, which can sometimes be inferred. Example 4 highlights the need for privacy researchers to be explicit about the data generating process, the adversary's assumptions, and what information one wishes to protect. This can be done using Pufferfish privacy and this is how we will make explicit the guarantees provided by differential privacy in our problem setting.

Given the data-generation model described in § 3.1, let $D_{1:T}$ be the sequence of sampled datasets, where $D_t = (X_{t,i})_{i \in L_t}$ denote the dataset produced at time $t$. Given a subset $S$ of $[T]$, we first define the boolean variable $\sigma_{(i,S)}$ that indicates whether an individual $i$'s data is present in $D_t$ for each $t \in S$ and not anywhere else:

$$\sigma_{(i,S)} := \left( \bigwedge_{t \in S} i \in L_t \right) \wedge \left( \bigwedge_{t \in [T] \setminus S} i \notin L_t \right).$$

We then define the secret pairs $\mathbb{Q}$ as follows, which state, in extensional form, that for each individual $i$, it is indistinguishable whether the person's status exists in a subset of the sampled datasets.

$$\mathbb{S} := \bigcup_{i \in [N^*]} \bigcup_{S \subseteq [T] : |S| \geq 1} \{\sigma_{(i,S)}\}$$

$$\mathbb{Q} := \bigcup_{i \in [N^*]} \bigcup_{S \subseteq [T] : |S| \geq 1} \bigcup_{R \subseteq S : |R| \geq 1} \{(\sigma_{(i,S)}, \sigma_{(i,S \setminus R)}\}$$

The additional element to specify in the Pufferfish framework is the data generating processes $\Theta$, representing the possible ways an attacker believes the data could have been generated. We define each $\theta \in \Theta$ to be a parameterization of the form $\theta := \{\mathscr{E}, \mu_{t,1}, \ldots, \mu_{t,N^*}\}_{t=1\ldots T}$, where $\mathscr{E}$ represents the underlying stochastic population process and is a distribution over the graph and status sequences, and $\mu_{t,i}$ is the probability that $i \in L_t$. We assume the attacker also has a distribution over the agent's policy that is integrated out in $\mathscr{E}$. Thus, we have

$$P(D_{1:T} \mid \theta) = \sum_{G_{1:T}} \sum_{X_{1:T}} \mathscr{E}(G_{1:T}, X_{1:T}) \prod_{t=1}^{T} P(D_t \mid X_t, \theta), \tag{3}$$

where

$$P(D_t \mid X_t, \theta) = \prod_{i \in L_t} \mu_{t,i} \prod_{j \neq L_t} (1 - \mu_{t,j}). \tag{4}$$

We do not place any restriction on $\mathscr{E}$, which models the attacker's prior belief over the likely sequence of interactions between individuals in the population and the attacker's knowledge about the dynamics model of the underlying population process.

The following result states the equivalence between $(\epsilon, \delta)$-Pufferfish privacy with parameters $(\mathbb{S}, \mathbb{Q}, \Theta)$ and $(\epsilon, \delta)$-differential privacy under $T$-fold adaptive composition (Dwork et al., 2010; Dwork & Roth, 2014). The full proof is given in Appendix A.

**Lemma 1.** A family of mechanisms $\mathcal{F}$ satisfies $(\epsilon, \delta)$-differential privacy under $T$-fold adaptive composition iff every sequence of mechanisms $\mathcal{M} = (\mathcal{M}_1, \ldots, \mathcal{M}_T)$, with $\mathcal{M}_i \in \mathcal{F}$, satisfies $(\epsilon, \delta)$-Pufferfish privacy with parameters $(\mathbb{S}, \mathbb{Q}, \Theta)$.

---

**Algorithm 1** Differentially Private Reinforcement Learning

---

1: **Input:** Environment $M = (\mathcal{S}, \mathcal{A}, r, P, \gamma)$, RL algorithm $\texttt{RL} : \mathcal{S} \times \mathcal{A} \times \mathcal{R} \times \mathcal{S} \times \Pi \to \Pi$, Privacy Mechanism $\mathcal{M} : \mathcal{S} \times \mathbb{R} \to \mathcal{S}$, initial state $s_0 \in \mathcal{S}$.
2: **Parameters** Privacy parameter $\epsilon'$, number of interactions $T$.
3: Randomly initialize policy $\pi_0$.
4: $\tilde{s}_0 = \mathcal{M}_{\epsilon'}(s_0)$.
5: **for** $t = 0, 1, 2, \ldots, T-1$ **do**
6:     $\tilde{a}_t \sim \pi_t(\cdot \,|\, \tilde{s}_t)$.
7:     Receive $s_{t+1} \sim P(\cdot \,|\, s_t, \tilde{a}_t)$.
8:     $\tilde{s}_{t+1} = \mathcal{M}_{\epsilon'}(s_{t+1})$.
9:     $\tilde{r}_t = r(\tilde{s}_{t+1}, \tilde{a}_t)$.
10:     $\pi_{t+1} \leftarrow \texttt{RL}((\tilde{s}_t, \tilde{a}_t, \tilde{r}_t, \tilde{s}_{t+1}), \pi_t)$.
11: **end for**

---

# 4 Differentially Private Reinforcement Learning

Our solution for differentially private reinforcement learning is presented in Algorithm 1. It is a meta algorithm that takes as input an MDP environment $M$, a reinforcement learning algorithm $\texttt{RL}$, a privacy mechanism $\mathcal{M}$, and parameters $(\epsilon, \delta)$ and $T$ that specify the level of privacy that should hold over $T$ interactions between the agent and the environment. The RL algorithm can be any method that takes a transition sample $(s, a, r', s')$ and a policy $\pi_{\text{old}}$ and outputs a new policy $\pi_{\text{new}} = \texttt{RL}((s, a, r', s'), \pi_{\text{old}})$. (Appendix C describes a concrete instantiation of Algorithm 1 using DQN as the RL algorithm.)

Our approach is to privatise the inputs before the RL algorithm receives them. We begin by first showing that Algorithm 1 satisfies the privacy guarantees specified in § 3.2. We then characterize the resulting control problem under our privacy approach before providing a utility bound.

## 4.1 Privacy Analysis

Algorithm 1 is constructed using differential privacy tools to satisfy $(\epsilon, \delta)$-differential privacy under $T$-fold adaptive composition. Since an individual's data is used at each time $t$ to output a state $s_t$, we need to privatise $s_t$ and ensure that all functions that take $s_t$ as input are also differentially private. Algorithm 1 does this by privatising every state using an $\epsilon'$-differentially private mechanism (lines 4 and 8). For instance, the Laplace mechanism with scale parameter $2/N\epsilon'$ (since $\Delta_q = 2/N$) would satisfy $\epsilon'$-differential privacy. As the state space is discrete and bounded, we also need to project the output from the Laplace mechanism back to the closest element in the state space. (See Algorithm 2 for more details.) The projection operation is guaranteed to be differentially private by Lemma 2.

**Lemma 2** (Post-processing (Dwork & Roth, 2014)). *Let $\mathcal{M} : \mathcal{X}^n \to Z$ be an $(\epsilon, \delta)$-differentially private mechanism and $f : Z \to Y$ an arbitrary function. Then the composition $f \circ \mathcal{M}$ is $(\epsilon, \delta)$-differentially private.*

In Algorithm 1, the action $\tilde{a}_t$ is selected using the current policy $\pi_t$ with the privatized state $\tilde{s}_t$ as input at every time step (line 6). Once the next state $s_{t+1}$ is sampled, it is immediately privatized to $\tilde{s}_{t+1}$ (lines 7 and 8). The agent then receives reward $\tilde{r}_t = r(\tilde{s}_{t+1}, \tilde{a}_t)$. Since the action $\tilde{a}_t$ and received reward $\tilde{r}_t$ are functions of the privatized state, they are also guaranteed private by Lemma 2. Thus, the entire transition sample received by the RL algorithm is $\epsilon'$-differentially private.

Lemma 3 now states the cumulative privacy guarantee over $T$ interactions.

**Lemma 3** (*T*-fold adaptive composition (Dwork et al., 2010; Dwork & Roth, 2014)). *For all $\epsilon', \delta', \delta \geq 0$, the class of $(\epsilon', \delta')$-differentially private mechanisms satisfies $(\epsilon, T\delta' + \delta)$-differential privacy under $T$-fold adaptive composition for:*

$$\epsilon = \sqrt{2T \log(1/\delta)}\epsilon' + T\epsilon'(e^{\epsilon'} - 1). \tag{5}$$

Combining Lemma 1, Lemma 2 and Lemma 3 then yields the following result.

**Theorem 1.** *Algorithm 1 satisfies $(\epsilon, \delta)$-differential privacy under T-fold adaptive composition, and equivalently $(\epsilon, \delta)$-Pufferfish privacy with parameters $(\mathbb{S}, \mathbb{Q}, \Theta)$ as defined in § 3.2.*

**Truthfulness in Data Collection**   A related question to privacy protection is an individual's willingness to honestly disclose her data to the data curator. For example, in the context of epidemic control, each individual $i$ may have a utility function $u_i : \mathbb{S} \to [0, 1]$ mapping states to a positive real number, with states that represent high infection rates assigned lower values because they are, perhaps, more likely to attract a mandated lock-down of the community. Would an individual who is sampled for data collection gain an advantage by misreporting her true infection status?

**Definition 4.** Given a (randomized) mechanism $\mathcal{M} : \mathbb{S} \to \mathbb{S}$, truthful reporting is an $\epsilon$-approximate dominant strategy for individual $i$ with utility $u_i$ and status $x_i$ if, for every dataset $D$ and every $y_i \neq x_i$,

$$\mathbb{E}_{o \sim \mathcal{M}(q(D \cup \{x_i\}))}\, u_i(o) \geq \mathbb{E}_{o \sim \mathcal{M}(q(D \cup \{y_i\}))}\, u_i(o) - \epsilon.$$

If truth reporting is an $\epsilon$-dominant strategy for every individual in the population, we say $\mathcal{M}$ is $\epsilon$-approximately dominant strategy truthful.

In McSherry & Talwar (2007), the authors show that any $(\epsilon, 0)$-differentially private mechanism is $\epsilon$-approximately dominant strategy truthful, which we can see by noting that

$$\mathbb{E}_{o \sim \mathcal{M}(q(D \cup \{x_i\}))}\, u_i(o) = \sum_o u_i(o) P(\mathcal{M}(q(D \cup \{x_i\})) = 0) \geq \sum_o u_i(o) e^{-\epsilon} P(\mathcal{M}(q(D \cup \{y_i\})) = 0)$$

$$= e^{-\epsilon}\, \mathbb{E}_{o \sim \mathcal{M}(q(D \cup \{y_i\}))}\, u_i(o) \geq \mathbb{E}_{o \sim \mathcal{M}(q(D \cup \{y_i\}))}\, u_i(o) - \epsilon.$$

The first inequality follows from Definition 1. The second inequality follows by noting that for $\epsilon < 1, e^{-\epsilon} \geq 1 - \epsilon$. Note that the argument does not work for $(\epsilon, \delta)$-differentially private mechanisms where $\delta > 0$. Algorithm 1 uses only an $(\epsilon, 0)$ differentially private mechanism in lines 4 and 8.

## 4.2   Utility Analysis

We now analyze the utility of our DPRL approach and present a theoretical result that bounds the approximation error of the optimal value function under privacy from the true optimal value function. The analysis we provide is asymptotic in nature and serves as an important step in establishing that good solutions are possible in our problem setting.

Whilst our approach to privacy in Algorithm 1 makes it easy to guarantee the differential privacy of any downstream RL algorithm, the learning and control problems are made more difficult as the true underlying process is now unobservable to the agent. Figure 2 visualizes the graphical model under our approach and highlights that the state, privatized states and actions evolve according to a partially-observable markov decision process (POMDP).

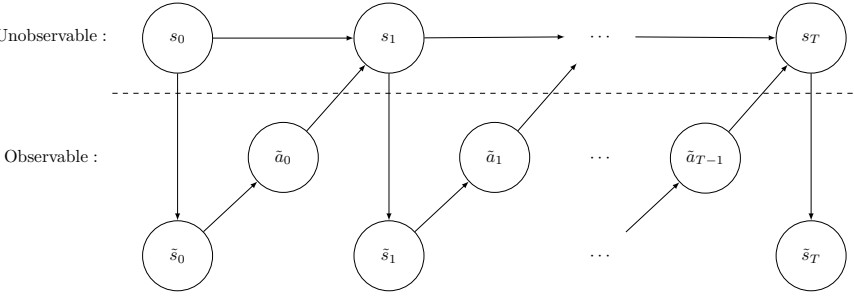

Figure 2: A graphical model of the underlying state and action sequence under our differentially private reinforcement learning approach. The true states are unobservable.

One subtle difference between a POMDP and our privatized system however is that the observed reward is not a direct function of the underlying state as that would constitute a privacy leak. The typical methods for solving POMDP problems resort to computationally expensive state-estimation techniques or sampling based approximations (Monahan, 1982; Shani et al., 2013; Kurniawati, 2022). Instead, we analyse the approximation error when standard MDP RL algorithms are applied directly on the observed privatized states without resorting to state-estimation. The analysis is done under several assumptions.

**Assumption 1** (Ergodicity)**.** The underlying MDP environment is ergodic (Puterman, 2014). An ergodic MDP ensures that a stationary distribution over states is well defined under any stationary policy.

Assumption 1 is a necessary tool when analyzing the asymptotic performance of RL algorithms and it is commonly used to ensure that every state-action pair is visited infinitely often (see e.g. Singh et al. (1994)).

**Assumption 2** (Lipschitz dynamics)**.** For all $s, s' \in \mathcal{S}, \ a \in \mathcal{A}$, there exists $L > 0$ such that

$$\|P(\cdot \,|\, s, a) - P(\cdot \,|\, s', a)\|_1 \le L \,\|s - s'\|_1 \,.$$

Since the privatized environment evolves as a POMDP, the distribution for the privatized state $\tilde{s}_t$ will in general depend on the entire history of observed states, actions and rewards. However, when an MDP RL algorithm is directly applied on top of privatized states, the transitions between privatized states are assumed to be Markovian. The induced transition model will however depend on the asymptotic state distribution under a behaviour policy generating interactions. For our analysis, we consider an off-policy setting where a stationary behaviour policy $\pi$ generates a sequence of privatized states, actions, and rewards. The induced Markovian transition model $\tilde{P}^\pi : \mathcal{S} \times \mathcal{A} \to \mathcal{D}(\mathcal{S})$ describes the asymptotic transition probabilities and is given by

$$\tilde{P}^\pi(\tilde{s}_{t+1} \,|\, \tilde{s}_t, \tilde{a}_t) = \sum_{s_t \in \mathcal{S}} \nu_\pi(s_t \,|\, \tilde{s}_t, \tilde{a}_t) \sum_{s_{t+1} \in \mathcal{S}} P(s_{t+1} \,|\, s_t, \tilde{a}_t) P_\mathcal{M}(\tilde{s}_{t+1} \,|\, s_{t+1}). \tag{6}$$

Here $P_\mathcal{M}(\tilde{s} \,|\, s) = \mathbb{P}(\mathcal{M}(s) = \tilde{s})$ denotes the distribution of the state privatization mechanism and $P(s_{t+1} \,|\, s_t, a_t)$ denotes the transition matrix of the underlying MDP. The transition model $\tilde{P}^\pi$ depends upon the behaviour policy $\pi$ through the distribution $\nu_\pi(s_t \,|\, \tilde{s}_t, \tilde{a}_t)$, which is the asymptotic probability of the underlying state being $s_t$ under $\pi$ when $\tilde{s}_t$ is observed and $\tilde{a}_t$ is performed. Using Bayes theorem, it can be expressed as

$$\begin{aligned}\nu_\pi(s \,|\, \tilde{s}, \tilde{a}) &= \frac{P_\mathcal{M}(\tilde{s} \,|\, s)\nu_\pi(s \,|\, \tilde{a})}{\sum_{s' \in \mathcal{S}} P_\mathcal{M}(\tilde{s} \,|\, s')\nu_\pi(s' \,|\, \tilde{a})} \\ &= \frac{P_\mathcal{M}(\tilde{s} \,|\, s)\nu_\pi(s \,|\, \tilde{a})}{\tilde{\nu}_\pi(\tilde{s} \,|\, \tilde{a})},\end{aligned} \tag{7}$$

where $\nu_\pi(s \,|\, \tilde{a})$ denotes the asymptotic probability of the underlying state being $s$ when $\tilde{a}$ is performed, and $\tilde{\nu}_\pi(\tilde{s}|\tilde{a}) = \sum_{s' \in \mathcal{S}} P_\mathcal{M}(\tilde{s} \,|\, s')\nu_\pi(s' \,|\, \tilde{a})$. To avoid degenerate cases, we will assume that the behaviour policy is stochastic and assigns non-zero probability to every action in all states.

**Assumption 3.** The behaviour policy $\pi$ is a stochastic policy where $\forall s \in \mathcal{S}, \forall a \in \mathcal{A}, \pi(a \,|\, s) > 0$.

Under Assumptions 1 and 3, the distributions $\nu_\pi(s \,|\, \tilde{a})$ and (7) are well defined.

Under our privatisation scheme, the induced MDP on top of privatized states is given by $\tilde{M} = (\mathcal{S}, \mathcal{A}, \tilde{P}^\pi, r, \gamma)$. Note that the reward function does not change as the received rewards are functions of the privatized states. For any policy $\bar{\pi} \in \Pi$, the Q-value $\tilde{Q}^{\bar{\pi}}$ in $\tilde{M}$ satisfies the Bellman equation $\tilde{Q}^{\bar{\pi}} = r + \gamma \tilde{P}^\pi \tilde{V}^{\bar{\pi}}$ where $\tilde{V}^{\bar{\pi}}(s) = \mathbb{E}_{a \sim \bar{\pi}(\cdot \,|\, s)}[\tilde{Q}^{\bar{\pi}}(s, a)]$.

Our analysis is performed using the projected laplace mechanism detailed in Algorithm 2. The projected laplace mechanism first applies additive noise to the input state as per the standard laplace mechanism. We require that the privacy mechanism we use has the state space as its output domain. Adding laplace noise no longer guarantees that the output will be in the state space, and so we take the orthogonal projection back onto the state space. This operation is guaranteed to be differentially private by Lemma 2.

The projected laplace mechanism satisfies the following tail bound. The proof is provided in Appendix B.

---
**Algorithm 2** Projected Laplace Mechanism

---
1: **Input:** state $s \in \mathcal{S}$
2: **Parameters:** Privacy parameter $\epsilon$, sensitivity parameter $\Delta_q$.

3: $s' = s + \eta$, where $\eta \in \mathbb{R}^K$ and for $i \in [K]$, $\eta_i \sim \text{Lap}\left(\frac{\Delta_q}{\epsilon}\right)$.
4: $\tilde{s} = \arg\min_{\bar{s} \in \mathcal{S}} \|s' - \bar{s}\|_2$.
5: **Return** $\tilde{s}$.

---

**Theorem 2.** *Let $X_t = (X_{t,i})_{i \in L_t}$ denote a dataset and let $s = q(X_t)$ and $\tilde{s} = \mathcal{M}(s)$, where $\mathcal{M}$ is the Projected Laplace mechanism. Then for all $\epsilon > 0$ and $\alpha > 0$,*

$$\mathbb{P}\left(\|s - \tilde{s}\|_\infty \geq \alpha + \frac{1}{\sqrt{2}N}\right) \leq K \exp\left(-\frac{N\alpha\epsilon}{2\sqrt{K}}\right).$$

Other privacy mechanisms were also considered but each have their own shortcomings. The exponential mechanism is a natural choice as it can be configured to output elements of the state space directly but the exponential mechanism is impractical to sample from. The output domains of additive noise mechanisms are typically unbounded and need to be modified for our problem setting. Whilst other approaches exist to bound the output to a particular domain, such as truncating the output (Holohan et al., 2020), we find working with the orthogonal projection simplifies the analysis. Another reason for working with the projected laplace mechanism is the fact that it can satisfy 'pure' differential privacy (i.e. $\delta = 0$). This is in contrast to mechanisms like the discrete or continuous gaussian mechanism (Dwork & Roth, 2014; Canonne et al., 2020) which only satisfy approximate differential privacy (i.e. $\delta > 0$). Since we use adaptive composition, satisfying approximate differential privacy at each time step would lead to the privacy budget increasing linearly with $T$ which is undesirable.

Our main utility result is the following bound on the approximation error between the optimal Q value in the underlying MDP $M$ and the privatized MDP $\tilde{M}$.

**Theorem 3.** *Let $M$ be the MDP environment and $\tilde{M}$ denote the privatised MDP under an $\epsilon$-differentially private projected laplace mechanism. Let $Q^*$ be the optimal value function in $M$ and $\tilde{Q}^*$ be the optimal value function in $\tilde{M}$. Then,*

$$\left\|Q^* - \tilde{Q}^*\right\|_\infty \leq \mathcal{O}\left(\sqrt{K}\exp\left(-\frac{\epsilon}{2\sqrt{2K}}\right) + K\exp\left(-\sqrt{N}\epsilon\right) + \frac{K^{\frac{3}{2}}}{\sqrt{N}}\right).$$

*Proof.* **(Sketch)** The full proof is given in Appendix B but we provide a sketch of the main ideas here.

The Simulation Lemma (Kearns & Singh, 2002) can be used to reduce the problem of bounding the Q-value error to the $L_1$ error $\|P_{sa} - \tilde{P}_{sa}^\pi\|_1$ between the transition models. Expanding the definition of $\tilde{P}_{sa}^\pi$ then allows us to split into two terms:

$$\|P_{sa} - \tilde{P}_{sa}^\pi\|_1 \leq \sum_{s_1 \in \mathcal{S}} \nu_\pi(s_1 \mid s, a)\left(\underbrace{\|P_{sa} - \bar{P}_{sa}\|_1}_{(one)} + \underbrace{\|\bar{P}_{sa} - \bar{P}_{s_1 a}\|_1}_{(two)}\right),$$

where $\bar{P}(s' \mid s_1, a) = \sum_{s_2 \in \mathcal{S}} P(s_2 \mid s_1, a)P_{\mathcal{M}}(s' \mid s_2)$. The first term can be viewed as the error due to privatising the output state from the transition model and the second term can be viewed as the error due to privatising the input state to the transition model.

The first term is bound by first applying the Bretagnolle-Huber inequality. The KL divergence between $P_{sa}$ and $\bar{P}_{sa}$ can be bound by noting that $\bar{P}_{sa}$ is a convolution between the privacy mechanism and the true transition model. The sum in the convolution can be reduced to a single element, leading to the bound:

$$\|P_{sa} - \bar{P}_{sa}^\pi\|_1 \leq 2\sqrt{1 - \min_{s'} P_{\mathcal{M}}(s'|s')},$$

where $P_{\mathcal{M}}(s'|s')$ denotes the probability of the projected laplace mechanism outputting $s'$ when the underlying state is $s'$. We can show $P_{\mathcal{M}}(s'|s') \geq 1 - K \exp\left(-\frac{\epsilon}{2\sqrt{2K}}\right)$, thus yielding a bound on term (one).

The second term is first bound by $\tilde{C} \sum_{s_1 \in \mathcal{S}} P_{\mathcal{M}}(s_1 \,|\, s) \left\|\bar{P}_{sa} - \bar{P}_{s_1 a}\right\|_1$, where $\tilde{C}$ is a constant. For $\alpha > 0$, let $B_\alpha(s) \coloneqq \{s' \in \mathcal{S} : \|s - s'\|_\infty < \alpha + 1/\sqrt{2}N\}$ be the $\ell_\infty$ ball centred on state $s$. The sum over $s_1$ is then split into $B_\alpha(s)$ and its complement $B_\alpha^c(s)$. The lipschitz property of our transition model and the concentration property of the projected laplace mechanism then allow us to bound the sum over $B_\alpha(s)$ and $B_\alpha^c(s)$ respectively. Term (two) is then bound as $\sum_{s_1 \in \mathcal{S}} \nu_\pi(s_1 \,|\, s, a) \left\|\bar{P}_{sa} - \bar{P}_{s_1 a}\right\|_1$ by $\mathcal{O}\left(2K \exp\left(-\frac{N\alpha\epsilon}{2\sqrt{K}}\right) + LK\alpha + \frac{LK}{\sqrt{2}N}\right)$. Choosing $\alpha = \frac{2\sqrt{K}}{\sqrt{N}}$ and combining terms then gives the final result. $\square$

Theorem 3 highlights how the approximation error scales as the population sample size $N$ and privacy parameter $\epsilon$ increase. For a given $K$, the approximation error decreases exponentially quickly as $\epsilon$ increases and at a rate of $N^{-\frac{1}{2}}$ as $N$ increases. Thus, increasing the sampled population size is an important factor in attaining good quality solutions for RL in population processes. Importantly, there are components in the upper bound that depend on only one of $N$ or $\epsilon$. This implies that both quantities must be increased to drive the error completely to zero. The bound also highlights that performance will degrade when the number of statuses of interest, $K$, increases. This is likely a function of the fact that the state space is discrete and scales exponentially with the dimension; its possible that formulating the problem with a query whose range is continuous and as a continuous reinforcement learning problem could avoid this issue. Nevertheless, our theoretical result shows the scaling behaviour of the approximation error and we corroborate this behaviour in our experiments.

## 5    Experiments

We present empirical results that corroborate our theoretical findings on the SEIRS Epidemic Control problem detailed in § 3. Our experiments on the Epidemic Control problem are representative of the class of population process environments as alternate problems will primarily differ in their transition dynamics. We encapsulate this by running experiments that vary the graph structure, population size, and transition parameters.

Table 1: Transition parameters for each experiment.

| Experiment | $\beta$ | $\sigma$ | $\gamma$ | $\rho$ |
|---|---|---|---|---|
| 1 | 0.3 | 0.5 | 0.143 | 0.015 |
| 2 | 0.5 | 0.1 | 0.15 | 0.01 |
| 3 | 0.2 | 0.3 | 0.1 | 0.01 |

We conduct three experiments. Each experiment corresponds to a different set of transition parameters for the SEIRS epidemic model (see Example 1) and the transition parameters used in each experiment are listed in Table 1. A graphical representation of the transition parameters and how they govern state transitions is shown in Figure 1. Every run of each experiment is conducted over $T = 2e5$ steps and the sample size taken to be 90% of the population size.

In each experiment the Epidemic Control problem is simulated over four large social networks from the Stanford Large Graph Network Dataset (Leskovec & Krevl, 2014). Table 2 lists the details of the datasets used, including the number of nodes and edges in each graph. These social networks represent reasonable models of social interactions in a population. The state space for each of these models is $\mathcal{O}(N^K)$, and that is computationally challenging for RL algorithms.

On each graph, the agent has access to 5 actions $\{\text{Quarantine}(i) : i \in \{0, 0.25, 0.5, 0.75, 1.0\}\}$, where $\text{Quarantine}(i)$ quarantines the top $i$th percent of nodes ranked by degree centrality by modifying the interaction matrix in the way described in § 3. The reward function is given by $r(s_t, a_t) = -(\alpha I(s_t) + (1 - \alpha)C(a_t))$

Table 2: Summary of network datasets used in experiments.

| Dataset | Name | Nodes | Edges |
|---|---|---|---|
| Slashdot (Leskovec et al., 2009) | 82K | 82,168 | 948,464 |
| Twitch (Rozemberczki & Sarkar, 2021) | 168k | 168,114 | 6,797,557 |
| Gowalla (Cho et al., 2011) | 196K | 196,591 | 950,327 |
| Youtube (Yang & Leskovec, 2012) | 1.1M | 1,134,890 | 2,987,624 |

and is taken as a convex combination between two functions $I(s_t)$ and $C(a_t)$. The function $I(s_t)$ returns the proportion of Exposed and Infected individuals at time $t$ and $C(a_t)$ returns the proportion of individuals quarantined by action $a_t$. We set $\alpha = 0.8$ in all experiments.

For each graph, we run simulations that vary the population size and the target cumulative privacy budget $\epsilon$ to see how the RL algorithm's performance changes as these key variables change. For given target cumulative privacy parameters $(\epsilon, \delta)$, we set the per-step privacy budget as

$$\epsilon' = f(\epsilon, \delta) = \frac{\epsilon}{2\sqrt{2T \log(1/\delta)}}, \tag{8}$$

and $\delta' = 0$. Under Lemma 3, setting the per-step privacy budget in this manner only satisfies the target value of $\epsilon$ for a certain range of values for $\delta$. As the value of $\delta$ decreases, the gap between the privacy achieved and the target privacy widens. In practice, $\epsilon$ values under 10 are of interest and we find that this can be achieved when $T = 2e5$ if $\delta \leq 10^{-2}$. For more details see Figure 5, Appendix C.

The RL algorithm we use is the DQN algorithm (Mnih et al., 2015) initialized with an experience replay buffer (Lin, 1992). We refer to its differentially private version as DP-DQN. DP-DQN can only store privatized transitions in its replay buffer. The DQN algorithm and the environment interact in an online fashion and exploration is performed using epsilon-greedy. The projected laplace mechanism was used as the state privatisation mechanism. A full description of the DP-DQN algorithm, a concrete algorithm for computing the projected laplace mechanism, and hyperparameters used is provided in Appendix C.

## 5.1 Results

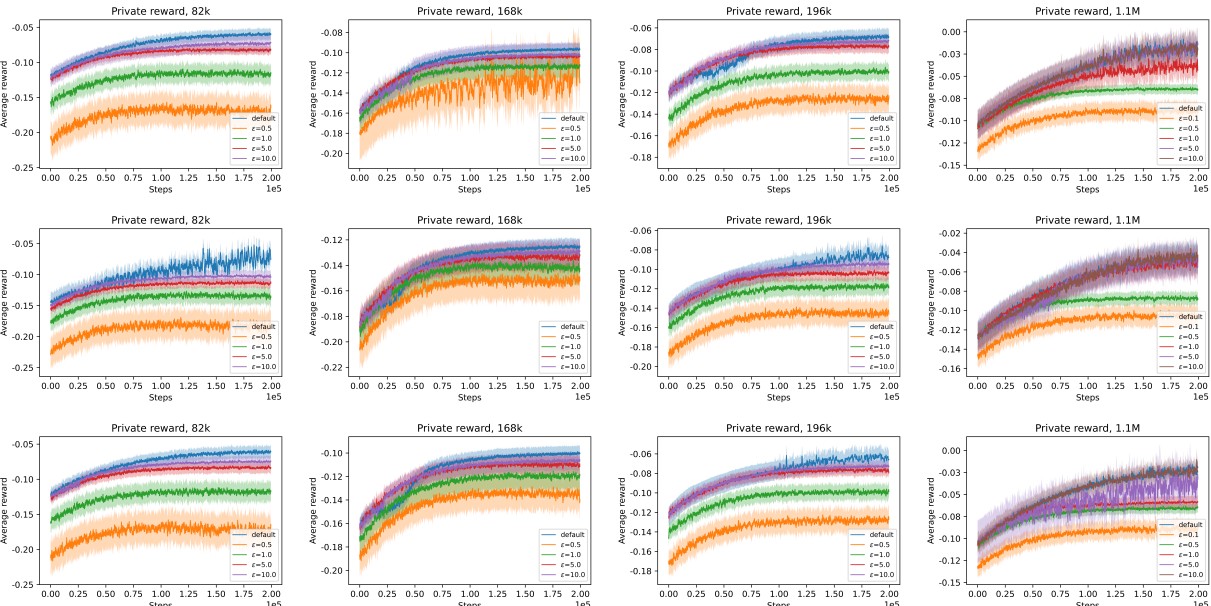

Figure 3: Private reward received by DP-DQN in Experiment 1 (top row), Experiment 2 (middle row), and Experiment 3 (bottom row).

Figures 3 and 4 plot the private reward and true reward performance of DP-DQN across all experiments. In each plot, we compare the performance of DP-DQN under differential privacy as $\epsilon$ is varied against the default performance of DP-DQN under no privacy. Each curve displays the mean and standard deviation over five random seeds. The privacy parameter $\delta$ was kept fixed across all runs at $\delta = 10^{-5}$. In all plots, the blue curve indicates the default performance of DP-DQN without differential privacy. Note that the blue curve does not change between the corresponding true reward and private reward plots.

The plots in Figure 3 show the performance of DP-DQN increases as $\epsilon$ increases. Notably, the performance clusters close to the default performance under low privacy (i.e. $\epsilon \geq 5$), and is indistinguishable from the default performance in the 1.1M graph. Additionally as the population size increases, the overall performance also improves. This falls in line with intuition as the noise added under a given privacy parameter has absolute scale and as the population size increases, the relative error due to privacy is much smaller. On the 1.1M graph we additionally plot the performance of DP-DQN under $\epsilon = 0.1$. Thus, the empirical results demonstrate in a finite sample setting the relationship between performance and the population size and privacy parameters highlighted by Theorem 3.

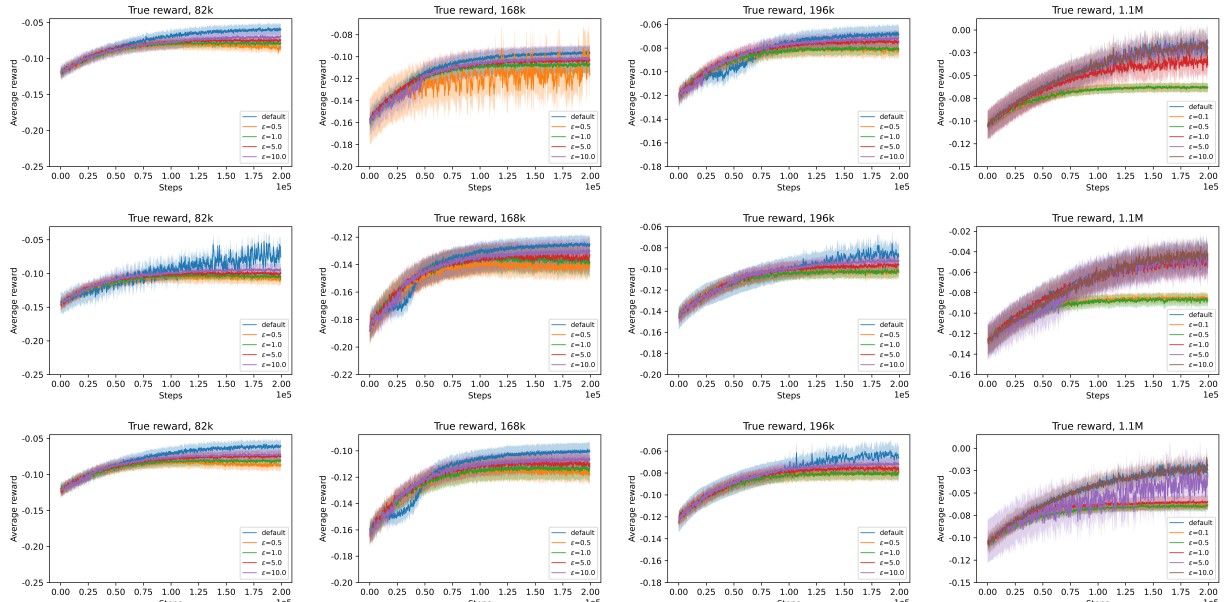

Figure 4: True reward received by DP-DQN in Experiment 1 (top row), Experiment 2 (middle row), and Experiment 3 (bottom row).

Whilst the results in Figure 3 are useful for corroborating our theoretical experiments, it is also important to investigate how the policy learnt under our privacy setup performs. Figure 4 plots the true reward received by DP-DQN across different graphs as $\epsilon$ varies. Overall, DP-DQN's performance increases as $\epsilon$ increases across all networks. Additionally, performance under privacy is clustered much closer to the default performance across all $\epsilon$ values in the 82k and 196k graphs than in the private reward case. The plots for the 1.1M graph show that performance under high privacy is indistinguishable from default performance. The gap in performance between the high privacy setting (i.e. $\epsilon \in \{0.1, 0.5\}$) and the low privacy setting on the 1.1M graph suggest that in some cases there may be a threshold value for $\epsilon$ that needs to be crossed to obtain almost optimal performance. Thus, we find that the policy learnt under our privacy setup can perform well and improves, drastically in some cases, as the privacy budget increases.

# 6    Limitations and Future Work

One limitation of our theoretical results is that they are asymptotic in nature. This analysis provides guarantees on the error between the solution under privacy and the true solution without privacy, but does not provide any guidance on whether such a solution can be learned. Nevertheless, our empirical results

provide some confidence the solutions found by learning algorithms will scale the way our results predict. Also, understanding the asymptotic properties of a problem setting is an important first step. Constructing and proving sublinear-regret algorithms in our problem setting is important future work. Additionally, whilst we have shown experimentally that the policy learnt under privacy can perform well, it would be useful to bound this performance theoretically and is the subject of further investigation.

Whilst our approach to achieving differential privacy is desirable as it is agnostic to the RL algorithm itself, it leaves open the possibility that the same privacy guarantees could be achieved by directly modifying an RL algorithm. Such an approach may be able to better deal with higher dimensional state spaces, which can adversely impact the approximation error, and is one of the limitations of our approach. Our analysis is also performed under some regularity assumptions; removing these assumptions would provide more generally applicable theoretical guarantees and is an interesting topic for future research.

## Impact Statement

As reinforcement learning algorithms see wider adoption and begin interacting with humans and their data, issues around protecting privacy become more pertinent. Our work can be seen as providing a promising start and solid theoretical foundation to providing privacy protections in an important problem class where reinforcement learning may be applied in the future.

## Acknowledgements

The authors would like to thank Mike Purcell for the many helpful discussions and whiteboard sessions during the early stages of the project.

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

## Appendix

## A    Proof of Lemma 1

**Lemma 1.** A family of mechanisms $\mathcal{F}$ satisfies $(\epsilon, \delta)$-differential privacy under $T$-fold adaptive composition iff every sequence of mechanisms $\mathcal{M} = (\mathcal{M}_1, \ldots, \mathcal{M}_T)$, with $\mathcal{M}_i \in \mathcal{F}$, satisfies $(\epsilon, \delta)$-Pufferfish privacy with parameters $(\mathbb{S}, \mathbb{Q}, \Theta)$ as defined in Section 3.2.

*Proof.* **We first show $T$-fold adaptive composition implies $(\epsilon, \delta)$-Pufferfish privacy.**

Suppose $\mathcal{M} = (\mathcal{M}_1, \ldots, \mathcal{M}_T)$ satisfies $(\epsilon, \delta)$ differential privacy under $T$-fold adaptive composition. Let $\mathfrak{D}_{1:T}$ be a random variable denoting the sequence of databases. Given an arbitrary $\theta$, secret $\sigma_{i,S}$, we have for each non-empty $R \subseteq S$,

$$P(\mathcal{M}(\mathfrak{D}_{1:T}) = y_{1:T} \,|\, \sigma_{(i,S)}, \theta)$$
$$= \sum_{G_{1:T}} \sum_{X_{1:T}} \sum_{D_{1:T} \,|\, \sigma_{(i,S)}} P(G_{1:T}, X_{1:T} \,|\, \theta) P(\mathfrak{D}_{1:T} = D_{1:T} \,|\, G_{1:T}, X_{1:T}, \sigma_{(i,S)}, \theta) P(\mathcal{M}(D_{1:T}) = y_{1:T})$$
$$\leq \delta + e^\epsilon \sum_{G_{1:T}} \sum_{X_{1:T}} \sum_{D_{1:T} \,|\, \sigma_{(i,S)}} P(G_{1:T}, X_{1:T} \,|\, \theta) P(\mathfrak{D}_{1:T} = D_{1:T} \,|\, G_{1:T}, X_{1:T}, \sigma_{(i,S)}, \theta) P(\mathcal{M}(D'_{1:T}) = y_{1:T}), \quad (9)$$

where $D'_{1:T}$ is obtained from $D_{1:T}$ by removing individual $i$'s data from $D_t$, $t \in R$. The inequality (9) follows from the property of T-fold adaptive composition:

$$\max_{\omega : P(\mathcal{M}(D_{1:T}) = \omega) \geq \delta} \ln \frac{P(\mathcal{M}(D_{1:T}) = \omega) - \delta}{P(\mathcal{M}(D'_{1:T}) = \omega)} \leq \epsilon. \quad (10)$$

Given a $D_{1:T}$ satisfying $\sigma_{(i,S)}$, we have

$$P(\mathfrak{D}_{1:T} = D_{1:T} \,|\, G_{1:T}, X_{1:T}, \sigma_{(i,S)}, \theta) = \left[ \prod_{t \in S} P(\mathfrak{D}_t = D_t \,|\, X_t, i \in L_t, \theta) \right] \left[ \prod_{t \in T \setminus S} P(\mathfrak{D}_t = D_t \,|\, X_t, i \notin L_t, \theta) \right].$$

Given $D'_{1:T}$ is obtained from $D_{1:T}$ as described above, it satisfies $\sigma_{(i,S \setminus R)}$ and we have

$$P(\mathfrak{D}_{1:T} = D'_{1:T} \,|\, G_{1:T}, X_{1:T}, \sigma_{(i,S \setminus R)}, \theta) =$$
$$\left[ \prod_{t \in R} P(\mathfrak{D}_t = D'_t \,|\, X_t, i \neq L_t, \theta) \right] \left[ \prod_{t \in S \setminus R} P(\mathfrak{D}_t = D_t \,|\, X_t, i \in L_t, \theta) \right] \left[ \prod_{t \in T \setminus S} P(\mathfrak{D}_t = D_t \,|\, X_t, i \notin L_t, \theta) \right].$$

For each $t \in R$, we have

$$P(\mathfrak{D}_t = D_t \,|\, X_t, i \in L_t, \theta) = P(\mathfrak{D}_t = D'_t \,|\, X_t, i \notin L_t, \theta) \quad (11)$$

since, by Equation (4), both the LHS and RHS of (11) are equal to

$$\prod_{j \in L_t \setminus \{i\}} \mu_{t,j} \prod_{j \in [N^*] \setminus L_t} (1 - \mu_{t,j}).$$

We have thus established that

$$P(\mathfrak{D}_{1:T} = D_{1:T} \,|\, G_{1:T}, X_{1:T}, \sigma_{(i,S)}, \theta) = P(\mathfrak{D}_{1:T} = D'_{1:T} \,|\, G_{1:T}, X_{1:T}, \sigma_{(i,S \setminus R)}, \theta). \quad (12)$$

Substituting Equation (12) into (9), and noting that the innermost summation $\sum_{D_{1:T} \,|\, \sigma_{(i,S)}}$ in (9) can be rewritten in the equivalent form $\sum_{D'_{1:T} \,|\, \sigma_{(i,S \setminus R)}}$, allows us to claim

$$P(\mathcal{M}(\mathfrak{D}_{1:T}) = y_{1:T} \,|\, \sigma_{(i,S)}, \theta) \leq \delta + e^\epsilon P(\mathcal{M}(\mathfrak{D}_{1:T}) = y_{1:T} \,|\, \sigma_{(i,S \setminus R)}, \theta).$$

Given $\theta \in \Theta, i \in [N^*], S \subseteq [T]$ and $R \subseteq S$ are all arbitrary, we have shown $\mathcal{M}$ satisfies $(\epsilon, \delta)$-Pufferfish privacy with parameters $(\mathbb{S}, \mathbb{Q}, \Theta)$.

**We next show $(\epsilon, \delta)$-Pufferfish privacy implies $T$-fold adaptive composition.**

Suppose $\mathcal{M} = (\mathcal{M}_1, \ldots, \mathcal{M}_T)$ satisfies $(\epsilon, \delta)$ Pufferfish privacy with parameters $(\mathbb{S}, \mathbb{Q}, \Theta)$. We show for any pair of neighbouring databases there is a $\theta \in \Theta$ that preserves differential privacy.

Let $D_{1:T}$ be an arbitrary sequence of datasets. For each individual $i$ in $D_{1:T}$, let $D'_{1:T}$ be obtained from $D_{1:T}$ by removing individual $i$'s data from one or more of the datasets. Thus, there exists $S$ and $R \subseteq S$ such that $D_{1:T}$ satisfies $\sigma_{(i,S)}$ and $D'_{1:T}$ satisfies $\sigma_{(i,S\setminus R)}$.

We choose $\theta = \{(\mathscr{E}, \mu_{t,1}, \ldots, \mu_{t,N^*}\}_{t=1..T}$ to be the following. $\mathscr{E}$ can be any stochastic population process. For each $t \in S$, define $\mu_{t,i} = 1/2$, and $\mu_{t,j} = 1$ for each $j \in L_t \setminus S$ and $\mu_{t,k} = 0$ for each $k \notin L_t$. And for each $t \notin S$, define $\mu_{t,j} = 1$ for each $j \in L_t$ and $\mu_{t,k} = 0$ for each $k \notin L_t$. Given $\mathcal{M}$ satisfies $(\epsilon, \delta)$ Pufferfish privacy, we have for any $y_{1:T}$:

$$P(\mathcal{M}(\mathfrak{D}_{1:T}) = y_{1:T} \,|\, \sigma_{(i,S)}, \theta) \leq e^\epsilon P(\mathcal{M}(\mathfrak{D}_{1:T}) = y_{1:T} \,|\, \sigma_{(i,S\setminus R)}, \theta) + \delta \qquad (13)$$

$$\Leftrightarrow P(\mathcal{M}(D_{1:T}) = y_{1:T} \,|\, \sigma_{(i,S)}, \theta) \leq e^\epsilon P(\mathcal{M}(D'_{1:T}) = y_{1:T} \,|\, \sigma_{(i,S\setminus R)}, \theta) + \delta \qquad (14)$$

$$\Leftrightarrow P(\mathcal{M}(D_{1:T}) = y_{1:T} \,|\, \theta) \leq e^\epsilon P(\mathcal{M}(D'_{1:T}) = y_{1:T} \,|\, \theta) + \delta \qquad (15)$$

Step (14) follows because of all the dataset sequences that can be generated using $\theta$, only $D_{1:T}$ satisfies $\sigma_{(i,S)}$ and only $D'_{1:T}$ satisfies $\sigma_{(i,S\setminus R)}$. Step (15) follows because $P(\sigma_{(i,S)} \,|\, \theta) = P(\sigma_{(i,S\setminus R)} \,|\, \theta) = (1/2)^{|S|}$.

Choosing $\theta \in \Theta$ in this manner for all neighbouring database sequences and repeating the calculations proves the final result. $\qquad \square$

In the proof of Lemma 1, we have only considered neighbouring datasets where an individual's data is dropped. The case when a neighbouring dataset $D'$ is obtained from a dataset $D$ by replacing an individual $i$'s data with another individual $j$'s data, where $j$ is not in $D$, can be formulated using the following secrets and secret pairs:

$$\sigma_{(i,S)} := \left( \bigwedge_{t \in S} i \in L_t \right) \wedge \left( \bigwedge_{t \in [T]\setminus S} i \notin L_t \right)$$

$$\sigma_{(i,S(W))} := \left( \bigwedge_{t \in S\setminus W_{|1}} i \in L_t \right) \wedge \left( \bigwedge_{t \in [T]\setminus(S\setminus W_{|1})} i \notin L_t \right) \wedge \left( \bigwedge_{(t,j) \in W} j \in L_t \right)$$

$$\mathbb{S}' := \bigcup_{i \in [N^*]} \bigcup_{S \subseteq [T]:|S| \geq 1} \{\sigma_{(i,S)}\}$$

$$\mathbb{Q}' := \bigcup_{i \in [N^*]} \bigcup_{S \subseteq [T]:|S| \geq 1} \bigcup_{\substack{W = \{(t,j)\,:\,t \in R, j \in [N^*]\setminus S\} \\ R \subseteq S \wedge |R| \geq 1}} \{(\sigma_{(i,S)}, \sigma_{(i,S(W))})\}$$

In the above, given $W = \{(t,x)\}$, we define $W_{|1} = \{t \,:\, \exists x.(t,x) \in W\}$. Using very similar arguments, one can show that $(\epsilon, \delta)$ differential privacy under T-fold adaptive composition is equivalent to $(\epsilon, \delta)$ Pufferfish privacy with parameters $(\mathbb{S}', \mathbb{Q}', \Theta)$.

For each of the secret pairs $(s_1, s_2) \in \mathbb{Q}'$, the Pufferfish privacy guarantees that

$$e^{-\epsilon} \leq \frac{P(s_1 \,|\, \mathcal{M}(\mathfrak{D}_{1:T}) = \omega, \theta)}{P(s_2 \,|\, \mathcal{M}(\mathfrak{D}_{1:T}) = \omega, \theta)} \bigg/ \frac{P(s_1 \,|\, \theta)}{P(s_2 \,|\, \theta)} \leq e^\epsilon. \qquad (16)$$

By inspecting the proof of Lemma 1, it is clear that secret pairs that represent an individual having two different status at a given time cannot be accommodated, and this is consistent with Example 4.

# B  Utility Analysis Proofs

## B.1  Laplace and Projected Laplace Mechanism Properties

The mechanism we consider using is the projected laplace mechanism. Denote by $\mathcal{M}_L$ the laplace mechanism which outputs $s' = \mathcal{M}_L(s) = s + (Y_1, \ldots, Y_K)$, $Y_i \sim Lap(\Delta q/\epsilon)$. The projected laplace mechanism outputs $\tilde{s} = \mathcal{M}(s)$ by first applying the laplace mechanism $s' = \mathcal{M}_L(s)$ and then takes the $\ell_2$ projection back onto the state space, i.e. $\tilde{s} = \arg\min_{\tilde{s}\in\mathcal{S}} \|\tilde{s} - s'\|_2$.

A concentration bound for the Laplace mechanism is given as follows.

**Theorem 4** (Dwork & Roth (2014)). *Let $X_t = (X_{t,i})_{i\in L_t}$ denote a dataset, $s = q(X_t)$ and $s' = \mathcal{M}_L(s)$. For $\epsilon > 0$ and $\delta \in (0,1]$,*

$$\mathbb{P}\left[\|s - s'\|_\infty \geq \ln\left(\frac{K}{\delta}\right)\cdot\left(\frac{\Delta q}{\epsilon}\right)\right] \leq \delta.$$

The following is a simple corollary of Theorem 4.

**Corollary 1.** Let $X_t = (X_{t,i})_{i\in L_t}$ denote a dataset and let $s = q(X_t)$ and $s' = \mathcal{M}_L(s)$. Then for $\epsilon > 0$ and $\alpha > 0$,

$$\mathbb{P}\left(\|s - s'\|_\infty \geq \alpha\right) \leq K\exp\left(-\frac{N\alpha\epsilon}{2}\right).$$

We now prove Theorem 2, a concentration bound for the projected laplace mechanism.

**Theorem 2.** Let $X_t = (X_{t,i})_{i\in L_t}$ denote a dataset and let $s = q(X_t)$ and $\tilde{s} = \mathcal{M}(s)$, where $\mathcal{M}$ is the Projected Laplace mechanism. Then for all $\epsilon > 0$ and $\alpha > 0$,

$$\mathbb{P}\left(\|s - \tilde{s}\|_\infty \geq \alpha + \frac{1}{\sqrt{2}N}\right) \leq K\exp\left(-\frac{N\alpha\epsilon}{2\sqrt{K}}\right).$$

*Proof.* First, define $P_{\Delta_K}(s) = \arg\min_{s'\in\Delta_K} \|s' - s\|_2$ and $P_{\mathcal{S}}(s) = \arg\min_{s'\in\mathcal{S}} \|s' - s\|_2$ as the $\ell_2$ projections onto the $K-1$ probability simplex $\Delta_K$ and the state space $\mathcal{S}$, noting that $\mathcal{S} \subset \Delta_K$. Letting $s' = \mathcal{M}_L(s)$, we can express $\tilde{s}$ as $\tilde{s} = \mathcal{M}(s) = P_{\mathcal{S}}(s')$.

Rather than finding an upper bound on $\mathbb{P}(\|s - \tilde{s}\|_\infty \geq \alpha)$, we instead find a lower bound on $\mathbb{P}(\|s - \tilde{s}\|_\infty < \alpha)$. Note that the $\ell_\infty$ ball of radius $\alpha$ (actually a hypercube) has greater volume than the $\ell_2$ ball of radius $\alpha$. Thus,

$$\mathbb{P}\left(\|s - \tilde{s}\|_\infty < \alpha\right) \geq \mathbb{P}\left(\|s - \tilde{s}\|_2 < \alpha\right). \tag{17}$$

Now define the following sets:

$$A := \left\{s' \in \mathbb{R}^K : \|s - P_{\mathcal{S}}(s')\|_2 < \alpha + \frac{1}{\sqrt{2}N}\right\}$$

$$B := \left\{s' \in \mathbb{R}^K : \|s - P_{\Delta_K}(s')\|_2 < \alpha\right\}$$

Note that $P_{\mathcal{S}}(s')$ must be a point that minimizes the $\ell_2$ distance to $P_{\Delta_K}(s')$, as $P_{\Delta_K}(s')$ is the minimum distance to the simplex $\Delta_K$ and minimizing the distance to $\mathcal{S}$ thus involves minimizing the distance along $\Delta_K$ from $P_{\Delta_K}(s')$. Also for two neighbouring points $s_1, s_2 \in \mathcal{S}$ (i.e. $s_1 \neq s_2$ and closest in $\ell_2$ distance), we must have two dimensions $i \neq j$ where $s_{1,i} = s_{2,i} - \frac{1}{N}$ and $s_{1,j} = s_{2,i} + \frac{1}{N}$ and $s_{1,k} = s_{2,k}$ for $k \neq i,j$. Thus the minimum distance between two neighbouring points is $\|s_1 - s_2\|_2 = \frac{\sqrt{2}}{N}$ and $P_{\mathcal{S}}(s')$ can be at most $\frac{1}{\sqrt{2}N}$ away from $P_{\Delta_K}(s')$. Then for $s' \in B$, we have:

$$\|s - P_{\mathcal{S}}(s')\|_2 \leq \|s - P_{\Delta_K}(s')\|_2 + \|P_{\Delta_K}(s') - P_{\mathcal{S}}(s')\|_2$$

$$\leq \alpha + \frac{1}{\sqrt{2}N}.$$

Thus $s' \in B \implies s' \in A$, implying $B \subseteq A$ and giving us $\mathbb{P}(A) \geq \mathbb{P}(B)$ or equivalently:

$$\mathbb{P}\left(\|s - P_{\mathcal{S}}(s')\|_2 < \alpha + \frac{1}{\sqrt{2N}}\right) \geq \mathbb{P}\left(\|s - P_{\Delta_K}(s')\|_2 < \alpha\right). \tag{18}$$

Let $\theta \in [0, \frac{\pi}{2}]$ denote the angle between $s - s'$ and the plane $\Delta_K$. Since $P_{\Delta_K}(s')$ is the orthogonal projection, we have $\|s - s'\|_2 = \|s - P_{\Delta_K}(s')\|_2 \cos^{-1}(\theta)$. Then we have:

$$\begin{aligned}
\mathbb{P}\left(\|s - P_{\Delta_K}(s')\|_2 < \alpha\right) &= \mathbb{P}\left(\|s - s'\|_2 < \alpha\cos^{-1}(\theta)\right) \\
&\overset{(a)}{\geq} \mathbb{P}\left(\|s - s'\|_\infty < \frac{\alpha\cos^{-1}(\theta)}{\sqrt{K}}\right) \\
&= 1 - \mathbb{P}\left(\|s - s'\|_\infty \geq \frac{\alpha\cos^{-1}(\theta)}{\sqrt{K}}\right) \\
&\overset{(b)}{\geq} 1 - K\exp\left(-\frac{N\alpha\epsilon\cos^{-1}(\theta)}{2\sqrt{K}}\right),
\end{aligned} \tag{19}$$

where (a) follows as $\{s' \in \mathbb{R}^K : \|s - s'\|_\infty \leq \alpha/\sqrt{K}\} \subseteq \{s' \in \mathbb{R}^K : \|s - s'\|_2 \leq \alpha\}$ – the former is a hypercube with side length $2\alpha/\sqrt{K}$ sitting completely inside the $\ell_2$ ball of radius $\alpha$ – and (b) follows by Corollary 1.

Combining (17), (18) and (19) then gives us:

$$\mathbb{P}\left(\|s - \tilde{s}\|_\infty < \alpha + \frac{1}{\sqrt{2N}}\right) \geq 1 - K\exp\left(-\frac{N\alpha\epsilon}{2\sqrt{K}}\right),$$

where we drop $\cos^{-1}(\theta)$ as $\cos^{-1}(\theta) \in [1, \infty]$ for $\theta \in [0, \frac{\pi}{2}]$. Thus, we have:

$$\begin{aligned}
\mathbb{P}\left(\|s - \tilde{s}\|_\infty \geq \alpha + \frac{1}{\sqrt{2N}}\right) &= 1 - \mathbb{P}\left(\|s - \tilde{s}\|_\infty < \alpha + \frac{1}{\sqrt{2N}}\right) \\
&\leq K\exp\left(-\frac{N\alpha\epsilon}{2\sqrt{K}}\right).
\end{aligned}$$

$\square$

## B.2 Additional Lemmas

The Simulation Lemma (Kearns & Singh, 2002; Agarwal et al., 2022) lets us bound the value function error in terms of the error in the transition functions.

**Lemma 4** (Simulation Lemma). Let $M = (\mathcal{S}, \mathcal{A}, r, P, \gamma)$ and $\tilde{M} = (\mathcal{S}, \mathcal{A}, r, \tilde{P}, \gamma)$ be two MDPs that differ only in the transition model. Given a policy $\pi$, let $Q^\pi$ be the value function under $\pi$ in $M$ and $\tilde{Q}^\pi$ be the value function under $\pi$ in $\tilde{M}$. Then for all $\pi$

$$\left\|Q^\pi - \tilde{Q}^\pi\right\|_\infty \leq \frac{\gamma}{1-\gamma}\left\|(P - \tilde{P})V^\pi\right\|_\infty.$$

**Lemma 5** (Lipschitz preservation under convolution). Suppose $f$ is an $L$-Lipschitz function and $\phi$ is a function such that $\int \phi(x)dx = 1$ and $\phi(x) \geq 0$ for all $x$. Then $g = f * \phi$ is also an $L$-Lipschitz function.

*Proof.*

$$\begin{aligned}
\|g(z) - g(z')\| &\leq \left\|\int (f(z + x) - f(z' + x))\phi(x)dx\right\| \\
&\leq \int \|f(z + x) - f(z' + x)\|\phi(x)dx \\
&\leq L\|z - z'\|\int \phi(x)dx \\
&= L\|z - z'\|.
\end{aligned}$$

$\square$

### B.3 Proof of Theorem 3

**Theorem 3.** Let $M$ be the MDP environment and $\tilde{M}$ denote the privatised MDP under an $\epsilon$-differentially private projected laplace mechanism. Let $Q^*$ be the optimal value function in $M$ and $\tilde{Q}^*$ be the optimal value function in $\tilde{M}$. Then,

$$\left\|Q^* - \tilde{Q}^*\right\|_\infty \leq \mathcal{O}\left(\sqrt{K}\exp\left(-\frac{\epsilon}{2\sqrt{2K}}\right) + K\exp\left(-\sqrt{N}\epsilon\right) + \frac{K^{\frac{3}{2}}}{\sqrt{N}}\right).$$

*Proof.* We have

$$\begin{aligned}
\left|Q^*(s,a) - \tilde{Q}^*(s,a)\right| &= \left|\sup_\pi Q^\pi(s,a) - \sup_\pi \tilde{Q}^\pi(s,a)\right| \\
&\leq \sup_\pi \left|Q^\pi(s,a) - \tilde{Q}^\pi(s,a)\right| \\
&\leq \sup_\pi \left\|Q^\pi - \tilde{Q}^\pi\right\|_\infty.
\end{aligned} \tag{20}$$

Then (20) can be bound with Lemma 4. For any policy $\bar{\pi}$ and any behaviour policy $\pi$ satisfying Assumption 3,

$$\begin{aligned}
\left\|Q^{\bar{\pi}} - \tilde{Q}^{\bar{\pi}}\right\|_\infty &\leq \frac{\gamma}{1-\gamma}\left\|(P - \tilde{P}^\pi)V^{\bar{\pi}}\right\|_\infty \\
&\leq \frac{\gamma}{1-\gamma}\max_{s,a}\left\|P_{sa} - \tilde{P}^\pi_{sa}\right\|_1 \left\|V^{\bar{\pi}}\right\|_\infty \\
&\leq \frac{\gamma r_{\max}}{(1-\gamma)^2}\max_{s,a}\left\|P_{sa} - \tilde{P}^\pi_{sa}\right\|_1.
\end{aligned} \tag{21}$$

In the above, $\tilde{P}^\pi$ is as defined in (6) and denotes the transition model in $\tilde{M}$ under behaviour policy $\pi$. Writing $\bar{P}(s' \,|\, s_1, a) = \sum_{s_2 \in \mathcal{S}} P(s_2 \,|\, s_1, a) P_\mathcal{M}(s' \,|\, s_2)$, we have, for any $s, a$,

$$\begin{aligned}
\left\|P_{sa} - \tilde{P}^\pi_{sa}\right\|_1 &= \left\|P_{sa} - \sum_{s_1 \in \mathcal{S}} \nu_\pi(s_1 \,|\, s, a)\bar{P}_{s_1 a}\right\|_1 \\
&= \left\|\sum_{s_1 \in \mathcal{S}} \nu_\pi(s_1 \,|\, s, a)P_{sa} - \sum_{s_1 \in \mathcal{S}} \nu_\pi(s_1 \,|\, s, a)\bar{P}_{s_1 a}\right\|_1 \\
&\leq \sum_{s_1 \in \mathcal{S}} \nu_\pi(s_1 \,|\, s, a)\left\|P_{sa} - \bar{P}_{s_1 a}\right\|_1 \\
&\leq \sum_{s_1 \in \mathcal{S}} \nu_\pi(s_1 \,|\, s, a)\left(\underbrace{\left\|P_{sa} - \bar{P}_{sa}\right\|_1}_{(one)} + \underbrace{\left\|\bar{P}_{sa} - \bar{P}_{s_1 a}\right\|_1}_{(two)}\right),
\end{aligned}$$

where the last two steps follows from the triangle inequality and splitting $\left\|P_{sa} - \bar{P}_{s_1 a}\right\|_1$. The first term can be viewed as the error due to output privatisation and the second term can be viewed as the error due to input privatisation. We will look to bound the two terms separately.

**Term (one)**
By the Bretagnolle-Huber inequality (Bretagnolle & Huber, 1979; Canonne, 2023), term (one) can be bound as:

$$\left\|P_{sa} - \bar{P}_{sa}\right\|_1 \leq 2\sqrt{1 - \exp(-D_{KL}(P_{sa} \,||\, \bar{P}_{sa})}.$$

The KL divergence term can be lower bound as follows:

$$
\begin{aligned}
-D_{KL}(P_{sa} \,\|\, \bar{P}_{sa}) &= \sum_{s' \in \mathcal{S}} P_{sa}^{s'} \log \frac{\bar{P}_{sa}^{s'}}{P_{sa}^{s'}} \\
&= \sum_{s' \in \mathcal{S}} P_{sa}^{s'} \log \frac{\sum_{s_2 \in \mathcal{S}} P_{\mathcal{M}}(s' \,|\, s_2) P_{sa}^{s_2}}{P_{sa}^{s'}} \\
&\geq \sum_{s' \in \mathcal{S}} P_{sa}^{s'} \log \frac{P_{\mathcal{M}}(s' \,|\, s') P_{sa}^{s'}}{P_{sa}^{s'}} \\
&= \sum_{s' \in \mathcal{S}} P_{sa}^{s'} \log P_{\mathcal{M}}(s' \,|\, s') \\
&\geq \min_{s' \in \mathcal{S}} \log P_{\mathcal{M}}(s' \,|\, s').
\end{aligned}
\tag{22}
$$

Here (22) follows as we shrink the sum to just when $s_2 = s'$.

Note that $P_{\mathcal{M}}(s' \,|\, s')$ is smallest when $s'$ is an interior point as it has the smallest subspace of $\mathbb{R}^K$ that projects back onto itself. So let $s' \in \mathcal{S} \setminus \partial\mathcal{S}$ denote an interior point going forward. Thus term (one) can be bound as:

$$
\left\| P_{sa} - \bar{P}_{sa} \right\|_1 \leq 2\sqrt{1 - P_{\mathcal{M}}(s' \,|\, s')}.
\tag{23}
$$

We now look to lower bound $P_{\mathcal{M}}(s' \,|\, s')$. Letting $\mathcal{R}(s')$ denote the subspace that projects onto $s'$, $P_{\mathcal{M}}(s' \,|\, s')$ can be expressed as

$$
\begin{aligned}
P_{\mathcal{M}}(s' \,|\, s') &= P_{\mathcal{M}_L}(s \in \mathcal{R}(s') \,|\, s') \\
&= \int_{s \in \mathcal{R}(s')} \left( \frac{N\epsilon}{4} \right)^K \exp\left( -\frac{N\epsilon}{2} \left\| s - s' \right\|_1 \right) ds.
\end{aligned}
$$

Let $\delta = \frac{1}{\sqrt{2N}}$ and $\mathcal{B}_\delta^2(s') \coloneqq \{ s \in \mathbb{R}^K : \left\| s - s' \right\|_2 \leq \delta \}$ denote the $\ell_2$ ball of radius $\delta$. Clearly, all points inside $\mathcal{B}_\delta^2(s')$ are closest to $s'$ compared to any other point in the state space. Also, we have that $\mathcal{B}_\delta^2(s') \subseteq \mathcal{R}(s')$. Integrating, we get:

$$
\begin{aligned}
P_{\mathcal{M}_L}(s \in \mathcal{R}(s') \,|\, s') &= \int_{s \in \mathcal{R}(s')} \left( \frac{N\epsilon}{4} \right)^K \exp\left( -\frac{N\epsilon}{2} \left\| s - s' \right\|_1 \right) ds \\
&\geq \int_{s \in \mathcal{B}_\delta^2(s')} \prod_{i=1}^K \left( \frac{N\epsilon}{4} \right) \exp\left( -\frac{N\epsilon}{2} |s_i - s_i'| \right) ds
\end{aligned}
$$

Let $\delta_K = \delta/\sqrt{K}$. We now integrate over $C_{\delta_K}(x') \coloneqq \bigtimes_{i=1}^K [s_i' - \delta_K, s_i' + \delta_K]$ to get a lower bound as $C_{\delta_K}(x') \subseteq \mathcal{B}_\delta^2(x')$. The integral is then:

$$
\begin{aligned}
P_{\mathcal{M}_L}(s \in \mathcal{R}(s') \,|\, s') &\geq \int_{s \in \mathcal{B}_\delta^2(s')} \prod_{i=1}^K \left( \frac{N\epsilon}{4} \right) \exp\left( -\frac{N\epsilon}{2} |s_i - s_i'| \right) ds \\
&\geq \int_{s \in C_{\delta_K}(s')} \prod_{i=1}^K \left( \frac{N\epsilon}{4} \right) \exp\left( -\frac{N\epsilon}{2} |s_i - s_i'| \right) ds \\
&= \prod_{i=1}^K \int_{s_i' - \delta_K}^{s_i' + \delta_K} \frac{N\epsilon}{4} \exp\left( -\frac{N\epsilon}{2} |s_i - s_i'| \right) ds_i \\
&= \prod_{i=1}^K \int_{-\delta_K}^{\delta_K} \frac{N\epsilon}{4} \exp\left( -\frac{N\epsilon}{2} |s_i| \right) ds_i
\end{aligned}
$$

$$= \prod_{i=1}^{K} \left(1 - \exp(-\alpha \delta_K)\right) \tag{24}$$

$$= \left(1 - \exp\left(-\frac{\epsilon}{2\sqrt{2K}}\right)\right)^K$$

$$\geq 1 - K \exp\left(-\frac{\epsilon}{2\sqrt{2K}}\right), \tag{25}$$

where step (24) follows by noting that for, all $\alpha > 0$

$$\frac{\alpha}{2} \int_{-\delta_K}^{\delta_K} \exp\left(-\alpha \left|x\right|\right) dx = \alpha \left(\int_0^{\delta_K} \exp\left(-\alpha x\right) dx\right) = \alpha \left[\frac{1 - \exp(-\alpha \delta_K)}{\alpha}\right] = 1 - \exp(-\alpha \delta_K)$$

and setting $\alpha = N\epsilon/2$, and step (25) follows by noting that for all $p \in [0, 1]$, $(1 - p)^k \geq 1 - kp$.

Substituting into equation (23) gives the following bound on term (one):

$$\sum_{s_1 \in \mathcal{S}} \nu_\pi(s_1 \,|\, s, a) \left\|P_{sa} - \bar{P}_{sa}\right\|_1 \leq 2\sqrt{1 - P_\mathcal{M}(s' \,|\, s')}$$

$$= 2\sqrt{1 - \left(1 - K \exp\left(-\frac{\epsilon}{2\sqrt{2K}}\right)\right)}$$

$$= 2\sqrt{K} \exp\left(-\frac{\epsilon}{2\sqrt{2K}}\right). \tag{26}$$

**Term (two)**
Using equation 7, we have:

$$\sum_{s_1 \in \mathcal{S}} \nu_\pi(s_1 \,|\, s, a) \left\|\bar{P}_{sa} - \bar{P}_{s_1 a}\right\|_1 = \sum_{s_1 \in \mathcal{S}} \frac{P_\mathcal{M}(s \,|\, s_1) \nu_\pi(s_1 \,|\, a)}{\tilde{\nu}_\pi(s \,|\, a)} \left\|\bar{P}_{sa} - \bar{P}_{s_1 a}\right\|_1$$

$$\leq H \sum_{s_1 \in \mathcal{S}} P_\mathcal{M}(s \,|\, s_1) \left\|\bar{P}_{sa} - \bar{P}_{s_1 a}\right\|_1$$

$$\leq H \sum_{s_1 \in \mathcal{S}} \frac{P_\mathcal{M}(s \,|\, s_1)}{P_\mathcal{M}(s_1 \,|\, s)} P_\mathcal{M}(s_1 \,|\, s) \left\|\bar{P}_{sa} - \bar{P}_{s_1 a}\right\|_1$$

$$\leq H C_{\max} \sum_{s_1 \in \mathcal{S}} P_\mathcal{M}(s_1 \,|\, s) \left\|\bar{P}_{sa} - \bar{P}_{s_1 a}\right\|_1,$$

where $H = \max_{s, s_1} \frac{\nu_\pi(s_1 \,|\, a)}{\tilde{\nu}_\pi(s \,|\, a)}$, and $C_{\max} = \max_{s, s_1} \frac{P_\mathcal{M}(s \,|\, s_1)}{P_\mathcal{M}(s_1 \,|\, s)}$.

We now look to bound $\sum_{s_1 \in \mathcal{S}} P_\mathcal{M}(s_1 \,|\, s) \left\|\bar{P}_{sa} - \bar{P}_{s_1 a}\right\|_1$. For $\alpha > 0$, define the $\ell_\infty$ ball of radius $\beta = \alpha + \frac{1}{\sqrt{2N}}$ around a state $s$ and its complement as:

$$B_\alpha(s) := \{s' \in \mathcal{S} : \|s - s'\|_\infty < \beta\}$$

$$B_\alpha^c(s) := \{s' \in \mathcal{S} : \|s - s'\|_\infty \geq \beta\}.$$

Splitting $\sum_{s_1 \in \mathcal{S}} P_{\mathcal{M}}(s_1 \mid s) \left\| \bar{P}_{sa} - \bar{P}_{s_1 a} \right\|_1$ over $B_\beta(s)$ and $B_\beta^c(s)$ gives us the following bound:

$$\sum_{s_1 \in \mathcal{S}} P_{\mathcal{M}}(s_1 \mid s) \left\| \bar{P}_{sa} - \bar{P}_{s_1 a} \right\|_1$$

$$= \sum_{s_1 \in B_\beta(s)} P_{\mathcal{M}}(s_1 \mid s) \left\| \bar{P}_{sa} - \bar{P}_{s_1 a} \right\|_1 + \sum_{s_1 \in B_\beta^c(s)} P_{\mathcal{M}}(s_1 \mid s) \left\| \bar{P}_{sa} - \bar{P}_{s_1 a} \right\|_1$$

$$\leq LK\beta + 2K \exp\left(-\frac{N\alpha\epsilon}{2\sqrt{K}}\right) \tag{27}$$

$$= 2K \exp\left(-\frac{N\alpha\epsilon}{2\sqrt{K}}\right) + LK\alpha + \frac{LK}{\sqrt{2}N}. \tag{28}$$

The first term in inequality (27) follows by noting that $\bar{P}$ is $L$-Lipschitz by Lemma 5 and Assumption 2, and that, for all $x \in \mathbb{R}^K$, $\|x\|_1 \leq K \|x\|_\infty$. The second term in inequality (27) follows by noting that the $L_1$ norm between distributions is bound by 2 and applying Proposition 2. Picking $\alpha = \frac{2\sqrt{K}}{\sqrt{N}}$ gives us

$$\sum_{s_1 \in \mathcal{S}} P_{\mathcal{M}}(s_1 \mid s) \left\| \bar{P}_{sa} - \bar{P}_{s_1 a} \right\|_1 \leq \mathcal{O}\left( K \exp\left(-\sqrt{N}\epsilon\right) + \frac{K^{3/2}}{\sqrt{N}} \right).$$

Thus the final bound on term (two) is:

$$\sum_{s_1 \in \mathcal{S}} \nu_\pi(s_1 \mid s, a) \left\| \bar{P}_{sa} - \bar{P}_{s_1 a} \right\|_1 \leq \mathcal{O}\left( K \exp\left(-\sqrt{N}\epsilon\right) + \frac{K^{3/2}}{\sqrt{N}} \right). \tag{29}$$

Combining (26) and (29) and picking the higher growth rate terms gives the final result. $\qquad\square$

## C    Implementation Details

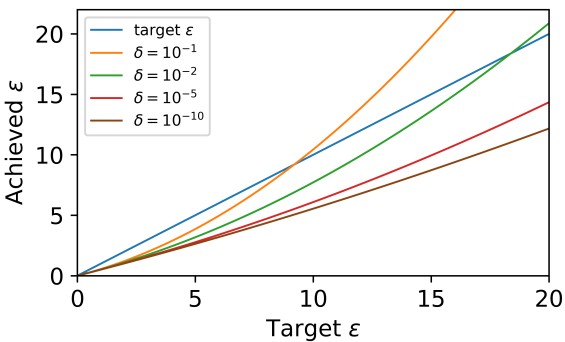

Figure 5: Target privacy vs privacy achieved under equations (5) and (8) as $\delta$ is varied. $T = 2e5$.

Figure 5 plots the curves for Equation 8 as a function of $\epsilon$ for different $\delta$ and T = 2e5 and shows that decreasing the value of $\delta$ increases the gap between the privacy achieved and the target privacy. Target $\epsilon$ values under 10 can be achieved for $\delta = 10^{-2}$.

Our experimental results use a concrete instantiation of Algorithm 1 with DQN (Mnih et al., 2015) and epsilon-greedy for exploration. We display the full algorithm details in Algorithm 4.

The state privatisation mechanism used is the projected laplace mechanism. We provide a particular instantiation of Algorithm 2 in Algorithm 3. Algorithm 3 works by first performing a projection onto the

simplex using the sorting method from Shalev-Shwartz et al. (2006) before finding the nearest point in the state space. The sorting method has complexity $\mathcal{O}(K \log K)$ and finding the nearest point in the state space can be done in $\mathcal{O}(K)$.

Algorithm 4 has a few minor differences to Algorithm 1. Firstly, the algorithm takes the target cumulative privacy parameters as input instead of the per-step privacy parameters. The per-step privacy budget is calculated in line 5. Lines 10-21 of can be considered the RL algorithm in Algorithm 1 expanded. The replay buffer and target network are written such that they are not a part of the RL algorithm itself. This is done to make clear that the buffer and target network maintain state across all iterations. Finally, DQN returns a value function instead of a policy. This is a minor change however as the policy used is simply the epsilon-greedy policy with respect to the returned value function.

The parameters used in each experiment are listed in Table 3. The neural network used in all experiments was a 6 layer, fully connected MLP. The learning rate ($\alpha$) is not listed as we use the default settings of the RMSProp optimizer in PyTorch to optimize the neural network.

All experiments were performed on a shared server with a 32-Core Intel(R) Xeon(R) Gold 5218 CPU, 192 gigabytes of RAM. A single NVIDIA GeForce RTX 3090 GPU was also used.

Table 3: DP-DQN parameters used for every experiment.

| $\gamma$ | $T$ | $B$ | $D$ | $\epsilon_{start}$ | $\kappa$ |
|---|---|---|---|---|---|
| 0.999 | 2e5 | 128 | 800 | 0.9999 | $10^{-5}$ |

---

**Algorithm 3** $\ell_2$-projected laplace mechanism

---

1: **Input:** state $s \in \mathcal{S} \subseteq \mathbb{R}^K$.
2: **Parameters:** Privacy parameter $\epsilon$, population size $N$.

3: $s' = s + \eta$, where $\eta \in \mathbb{R}^K$ and for $i \in [K]$, $\eta_i \sim \text{Lap}\left(\frac{2}{N\epsilon}\right)$.
4: Sort $s' = (s'_1, \ldots, s'_K)$ in descending order: $s'_{(1)} \leq \ldots \leq s'_{(n)}$.
5: Define $\pi(m) = \frac{1}{m}\left(\sum_{r=1}^{m} s'_{(r)} - 1\right)$
6: Compute $\rho = \max\left\{m \in [K] : s'_{(m)} - \pi(m) > 0\right\}$
7: Compute $\bar{s}$ where $\bar{s}_i = \max(0, s'_i - \pi(\rho))$
8: Compute $e \in \mathbb{R}^K$ where $e_i = x_i - \bar{s}_i$ and $x_i = \arg\min_{x \in S_1} |x - \bar{s}_i|$ for $S_1 = \left\{0, \frac{1}{N}, \ldots, \frac{N-1}{N}, 1\right\}$.
9: Find $J \subset [K]$ with $|J| = K - 1$ that minimizes $\sqrt{\sum_{i \in J} e_i^2}$. Let $j = [K] \setminus J$ and set $e_j = -\sum_{i \in J} e_i$.
10: **return** $\bar{s} + e$.

---

---

**Algorithm 4** Differentially Private DQN (DP-DQN)

---

1: **Input:** Environment $M = (\mathcal{S}, \mathcal{A}, r, P, \gamma)$, initial state $s_0 \in \mathcal{S}$, privacy mechanism $\mathcal{M} : \mathcal{S} \times \mathbb{R} \to \mathcal{S}$.
2: **Parameters:** cumulative privacy parameters $(\epsilon, \delta)$, time horizon $T$, batch size $B$, target update step $D$, population size $N$, learning rate $\alpha$, initial exploration rate $\epsilon_{start} < 1$, decay rate $\kappa < 1$.
3: **Initialize:** network parameters $\theta$ randomly, target network parameters $\bar{\theta} = \theta$, $\epsilon_{explore} = \epsilon_{start}$.
4: `Buffer` $\leftarrow \{\}$.
5: $\epsilon' = \frac{\epsilon}{2\sqrt{2T \log(1/\delta)}}$.
6: $\tilde{s}_0 = \mathcal{M}_{\epsilon'}(s_0)$.
7: **for** $t = 0, 1, 2, \ldots, T$ **do**
8: $\quad$ $p \sim Uniform([0, 1])$.
9: $\quad$ **if** $p > \epsilon_{explore}$ **then** $\tilde{a}_t = \arg\max_a Q_\theta(\tilde{s}_t, a)$ **else** $\tilde{a}_t \sim Uniform(\mathcal{A})$.
10: $\quad$ Receive $s_{t+1} \sim P(\cdot \,|\, s_t, \tilde{a}_t)$.
11: $\quad$ $\tilde{s}_{t+1} = \mathcal{M}_{\epsilon'}(s_{t+1})$.
12: $\quad$ $\tilde{r}_t = r(\tilde{s}_t, a_t)$,
13: $\quad$ Append $(\tilde{s}_t, \tilde{a}_t, \tilde{r}_t, \tilde{s}_{t+1})$ to `Buffer`.
14: $\quad$ **if** $t > B$ **then**
15: $\quad\quad$ **for** $i = 1, \ldots, B$ **do**
16: $\quad\quad\quad$ $(s, a, r, s') \sim Uniform(\texttt{Buffer})$.
17: $\quad\quad\quad$ $y_i \leftarrow r + \gamma \max_a Q_{\bar{\theta}}(s', a)$.
18: $\quad\quad\quad$ $\ell_i \leftarrow \frac{1}{2}\left(Q_\theta(s, a) - y_i\right)^2$.
19: $\quad\quad$ **end for**
20: $\quad\quad$ Run one step SGD $\theta \leftarrow \theta + \alpha \frac{1}{B} \nabla_\theta \sum_{j=1}^{B} \ell_j$.
21: $\quad$ **end if**
22: $\quad$ **if** $t \mod D = 0$ **then**
23: $\quad\quad$ Update target network parameters $\bar{\theta} \leftarrow \theta$.
24: $\quad$ **end if**
25: $\quad$ $\epsilon_{explore} \leftarrow 0.03 + (\epsilon_{start} - 0.03) \cdot e^{-\kappa t}$.
26: **end for**

---

