# OpenReview forum: "Privacy Preserving Reinforcement Learning for Population Processes"
_TMLR — Accepted by TMLR_

### Review · Reviewer_K4Lb · 2024-07-18

**Summary Of Contributions:**

This paper proposes a theory that reinforcement learning can be introduced with privacy-preserving for population processes, and experiments validate this idea.

**Audience:**

Yes

**Broader Impact Concerns:**

I do not see any broader impact concerns.

**Claims And Evidence:**

Yes

**Requested Changes:**

1. Please add some newer dataset.
2. Add some comparisons with other works.

**Strengths And Weaknesses:**

Strengths:

Extensive experiments based on different parameters.
Clear explanations of basic concepts.

Weaknesses:

Short of using the latest datasets, the latest dataset that this paper uses is 12 years old, and there are some newer datasets available.
Lack of comparison with other similar works that use DP and RL

---

> ### Author Response · Authors · 2024-08-17
>
> Thank you for the review.
>
> - _"1. Please add some newer dataset."_
>
> Whilst we are willing to perform experiments on a newer graph dataset, we question the value added in doing so. The main consideration when varying the graph datasets used is the change in graph structure and population size (i.e. number of nodes). Additionally the graphs we used all reflect social networks and it is unclear whether social patterns have substantially changed in the past 12 years. If the reviewer believes that a newer dataset offers additional insight over what we have provided, we would be willing to run the experiments on a newer graph dataset.
>
> - _"2. Add some comparisons with other works."_
>
> As we mention in our related work section, comparison to other DPRL approaches do not make a lot of sense as they are targeted at different privacy settings. A more detailed discussion of an example may help elucidate this point. In [1], a dataset consists of $m$ trajectories and a neighbouring dataset differs in the last trajectory. In our setting, an individual can exist across all trajectories, and so the privacy guarantees of [1] don't apply. To adapt their approach to our setting, the definition of neighbouring database would have to be adjusted to consider all databases as neighbours, and consequently the variance of the Gaussian noise that needs to be added would be too large.
>
> Our results do however compare against the performance under no privacy. This can be regarded as a sensible baseline to compare to and is typically the type of experimental analysis performed in DPRL when no reasonable comparative algorithms exist (see [1], [2]).
>
> [1] Borja Balle, Maziar Gomrokchi, and Doina Precup. Differentially private policy evaluation, 2016.
>
> [2] Baoxiang Wang and Nidhi Hegde. Privacy-preserving q-learning with functional noise in continuous spaces.
> Advances in Neural Information Processing Systems, 32, 2019.

---

> > ### Comment · Reviewer_K4Lb · 2024-08-30
> > **Regarding new dataset**
> >
> > I would suggest using some newer dataset; here are some reasons:
> >
> > While the structure of social networks might not have changed dramatically, newer datasets could include more recent trends or patterns in social behavior that older datasets do not capture. For example, social media usage, interaction patterns, and network formations have evolved, especially with the rise of different platforms.
> >
> > Using outdated datasets could potentially limit the generalizability of their findings to current real-world applications. A newer dataset could provide additional validity to their results and make their findings more applicable to modern social networks.
> >
> > Technological advances, changes in how people interact online, and the proliferation of different platforms (e.g., the rise of TikTok or changes in Facebook’s user base) could influence network structures in ways that are not captured by older datasets.

---

> > > ### Author Response · Authors · 2024-08-30
> > >
> > > We will begin running experiments on a newer dataset immediately. We have chosen the Twitch social network dataset from [1] as it is more recent and of sufficient size.
> > >
> > > [1] Benedek Rozemberczki and Rik Sarkar. Twitch Gamers: a Dataset for Evaluating Proximity Preserving and Structural Role-based Node Embeddings, 2021

---

> > > > ### Author Response · Authors · 2024-09-03
> > > > **Results on a newer dataset added**
> > > >
> > > > We have updated the manuscript to include results on the Twitch social network dataset. Please see both Figures 3 and 4 in the revised manuscript.

---

### Review · Reviewer_x386 · 2024-08-11

**Summary Of Contributions:**

The paper considers privacy preserving reinforcement learning where the state space consists of some aggregation of the nodes of an evolving graph. The actions influence the edges of the graph, which in turn influence the evolution of the state of the nodes. The paper advocates the use of Pufferfish privacy that seems to capture better the correlated data in the population than differential privacy. A differentially private reinforcement learning algorithm is proposed that treats the (online) reinforcement learning algorithm as black box. The algorithm induces privacy by adding noise to the state (i.e. the aggregated statistics). The algorithm is shown to achieve differential privacy and Pufferfish privacy. Moreover, an asymptotic bound is proven for the drop in the optimal Q/value depending on the privacy level. Empirical analysis on some standard networks using DQN as learning component demonstrate the behavior of the privacy preserving algorithm, showing that the average reward does indeed depend on the privacy.

**Audience:**

Yes

**Claims And Evidence:**

No

**Requested Changes:**

Even if the reward is computed by the agent, the theoretical analyses and the experimental evaluations should be in the realm of the true reward function.

The presentation of the setting implies that the agent actions have direct influence on the edges of the graph, e.g. quarantining an individual. That would seem to be a clear breach of privacy (the agent should not know the state of a node). I could understand a more indirect influence, like changing the quarantining rules, which if acted upon by the trusted owner of the data could indeed lead to limiting the edges of a node. This issue should be clarified by the authors.

I the intended RL setting is online, then this issue should be highlighted and the appropriate theoretical discussion should be provided.

The experiments do not include any baselines. I understand that this is a fairly different setting than the existent private RL dealt with, but perhaps more effort could have been made to include some baseline that are possible to adapt to the setting at hand.

**Strengths And Weaknesses:**

The problem seems relevant and it provides an interesting insight on differential and Pufferfish privacy for population processes.

The proposed algorithm is reasonable, but there is no significantly new idea in it.

My main concern regards the reward function. The reward function is computed by the learning agent, which is already unorthodox, since the reward is usually provided by the environment in reinforcement learning. This choice is less problematic though. The bigger problem is that reward depends on the privatized state. Given that the reward is computed by the learning agent, which is not trusted, this seems a natural choice. However, because the bound on the optimal Q-value considers the difference between the optimal Q-value corresponding to the 'true' states and the the optimal Q-value corresponding to the 'privatized' states, the bound becomes meaningless. It is quite easy to see that it is possible to design a privatization function that maximizes the reward function while giving no information out (the state becomes a constant vector that achieves the highest regret). It would probably better to bound the difference between the optimal Q-value in the original MDP and the expected 'true' reward achieved by the policy that is optimal in the privatized MDP.

Another issue is that the setting is one of online reinforcement learning, where one would be expected to analyze the regret of the algorithm, i.e. a bound on the cumulative reward.

The experiment suffer the same problem that it is comparing average reward corresponding to different reward functions, therefore an algorithm might achieve higher average reward because it is performing better, but also because it works with a more favorable reward function.

---

> ### Author Response · Authors · 2024-08-17
> **Response part 1**
>
> We would like to thank the reviewer for their thoughtful comments. We address the concerns stated below.
>
> - _"The reward function is computed by the learning agent, which is already unorthodox, since the reward is usually provided by the environment in reinforcement learning."_
>
> We apologize for the misleading statement in section 3.1, step 4 which states that the agent computes the rewards and forms the reward sample. What we actually mean, and what is stated in the rest of the paper, is for the reward function to be computed by the Data Curator, who is external to the learning agent and part of the environment. From Algorithm 1, the learning agent is identified with the RL algorithm (step 10) and does not compute the reward. We will rewrite the relevant parts of section 3.1 to rectify this issue.
>
> - _"However, because the bound on the optimal Q-value considers the difference between the optimal Q-value corresponding to the 'true' states and the the optimal Q-value corresponding to the 'privatized' states, the bound becomes meaningless. It is quite easy to see that it is possible to design a privatization function that maximizes the reward function while giving no information out (the state becomes a constant vector that achieves the highest regret). It would probably better to bound the difference between the optimal Q-value in the original MDP and the expected 'true' reward achieved by the policy that is optimal in the privatized MDP."_
>
> We disagree with the statement that our bound is meaningless. Our bound provides a guarantee that the optimal Q value in the privatised MDP will converge to the optimal Q-value in the true MDP as the privacy budget and population size increase.
> A corollary of our bound is that the optimal policy in the privatised MDP will converge to the optimal policy in the original MDP, which is a guarantee on the expected `true' reward achieved asymptotically. This can be seen as follows.
>
> Let $Q^*$ denote the optimal value function in the original MDP and $\tilde{Q}^*$ denote the optimal value function in the privatised MDP. Let $A(s,a) = \{a' \in A: Q^*(s, a') = Q^*(s, a) \}$ denote the set of actions that achieve the same value as $a$ in state $s$. Let $\Delta(s, a) = \min_{a' \in A \setminus A(s, a)} |Q^*(s, a) - Q^*(s, a')|$ denote the gap in value between an optimal action and a next best action. Our result implies that there exists $\epsilon, N < \infty$ where $|Q^*(s, a) - \tilde{Q}(s, a)| < \Delta / 2$. Beyond this point, the optimal policy in the privatised MDP will perform exactly as the optimal policy in the original MDP asymptotically. Our result also implies that the error will shrink exponentially quickly as $\epsilon$ and $N$ increase.
>
> The example privatised function you propose is also incompatible with our results. Firstly, Theorem 3 is proven using the projected laplace mechanism; proving bounds using a different privatisation function requires a different analysis. Additionally, the privatised function you propose cannot achieve the bound in our results. Our bound is on the $\ell_\infty$ norm, which bounds the error on every term of the value function. If the privatisation function only outputs to a single state or a subset of the state space, the error would be unbounded / undefined. More generally, a method that achieves linear regret independent of $\epsilon$ and $N$ cannot attain our asymptotic bound as the error does not go to 0 under linear regret.
>
> Ultimately, whilst we do acknowledge that fixed-time guarantees would be more useful, something we acknowledge as being important future work, this does not negate our claims or their utility.
>
> - _"The experiment suffer the same problem that it is comparing average reward corresponding to different reward functions..."_
>
> Our current experimental results do provide empirical evidence for our theoretical result, showing in practice that the average reward (a proxy for the value function) will converge to the performance under no privacy. We acknowledge that also plotting against the true reward would be useful and we will conduct these experiments. As the experiments will take some time, we aim to have the updated manuscript uploaded by the end of the rebuttal period.

---

> > ### Author Response · Authors · 2024-08-17
> > **part 2**
> >
> > - _"The experiments do not include any baselines. I understand that this is a fairly different setting than the existent private RL dealt with, but perhaps more effort could have been made to include some baseline that are possible to adapt to the setting at hand."_
> >
> > As the noise added to protect privacy is always tailored to the specific privacy setting, we did not find any DPRL approaches that were readily adaptable to our setting. A more detailed discussion of an example may help elucidate this point. In [1], a dataset consists of $m$ trajectories and a neighbouring dataset differs in the last trajectory. In our setting, an individual can exist across all trajectories, and so the privacy guarantees of [1] don't apply. To adapt their approach to our setting, the definition of neighbouring database would have to be adjusted to consider all databases as neighbours, and consequently the variance of the Gaussian noise that needs to be added would be too large.
> >
> > Our results do however compare against the performance under no privacy. This can be regarded as a sensible baseline to compare to and is typically the type of experimental analysis performed in DPRL when no reasonable comparative algorithms exist (see [1], [2]).
> >
> > [1] Borja Balle, Maziar Gomrokchi, and Doina Precup. Differentially private policy evaluation, 2016.
> >
> > [2] Baoxiang Wang and Nidhi Hegde. Privacy-preserving q-learning with functional noise in continuous spaces.
> > Advances in Neural Information Processing Systems, 32, 2019.
> >
> >
> > - _"The presentation of the setting implies that the agent actions have direct influence on the edges of the graph, e.g. quarantining an individual. That would seem to be a clear breach of privacy (the agent should not know the state of a node). I could understand a more indirect influence, like changing the quarantining rules, which if acted upon by the trusted owner of the data could indeed lead to limiting the edges of a node. This issue should be clarified by the authors._"
> >
> > Whilst the action space is indeed made up of different quarantining rules, these are determined and executed by the environment. The agent is unable to influence, let alone see, any one specific individual in the population except through the population-level quarantining rules it selects.
> >
> > Ultimately, the concerns raised here do not fall under the guarantees of Differential Privacy. As we discuss in Section 3.2, one has to be wary of what one expects Differential Privacy to protect. The following example adapted from [3] highlights how 'harm' can be caused even though Differential Privacy is guaranteed:
> >
> > Consider an insurance assessor, Alice, who possesses prior knowledge that a customer, John, regularly smokes. If a clinical study reports a meaningful causal relationship between smoking and lung disease, it would lead Alice to conclude that John has a higher risk of lung disease compared to non-smokers and therefore should have his insurance premium increased. This occurs regardless of whether John actually participated in the clinical study. What DP provides in this case is only that John’s insurance premium increase is *not* due to the presence of his data in the clinical study.
> >
> > The potentially confusing semantics of DP is a fundamental reason why we explicitly state our privacy protections using Pufferfish privacy.
> >
> > [3] S.P. Kasiviswanathan, A. Smith, On the `Semantics’ of Differential Privacy: A Bayesian Formulation, Journal of Privacy and Confidentiality

---

> ### Author Response · Authors · 2024-09-03
> **True reward results added**
>
> We have updated our paper to include additional experimental results comparing the true reward performance of DP-DQN under privacy. Please see Figure 4 in the revised manuscript.

---

### Review · Reviewer_1uT5 · 2024-08-15

**Summary Of Contributions:**

The authors consider the problem of differential privacy in RL algorithms on population processes and provide a meta-algorithm with asymptotic guarantees. The authors provide empirical results that demonstrate the effectiveness of this method on 3 real graphs which gives credibility to their claim that the algorithm is useful in fixed time.

**Audience:**

Yes

**Claims And Evidence:**

Yes

**Requested Changes:**

Since the technical analysis does not offer any fixed-time guarantees, I believe that the empirical results need to be a bit more extensive. The presentation of the empirical results is clear and details are provided, but in its current form, I only have confidence in the main result holding for the SEIRS problem. Empirical evaluation on other problems would give more confidence in this result holding elsewhere. Alternatively, if the SEIRS problem encompasses most population processes this should be more clearly stated.

**Strengths And Weaknesses:**

Strengths
- The paper is written very well. The introduction, relevant works, and technical preliminaries are complete and concise.
- The meta-algorithm is presented in full generality, and sufficient detail is provided in the empirical results for the reader to understand a tangible instantiation of the method.
- The technical analysis is thorough and relevant proofs are clear. The analysis appears to be correct on initial inspection.

Weaknesses
- The asymptotic result requires both $\epsilon$ and $N$ to grow large. This asymptotic result would be much stronger if this result would hold when only $N$ grows large. Although I am not an expert in DP, this seems like a weak result because of this.
- The empirical evaluation is limited in the number of problems considered. Specifically, only the SEIRS problem is considered on 3 graphs.

---

> ### Author Response · Authors · 2024-08-17
>
> We thank the reviewer for their comments. In regards to your requested changes:
>
> The SEIRS problem we consider is representative of many different population process problems and we agree that we should state this more clearly.
> The free parameters in population processes are the population size $N$, graph structure (given by the underlying contact network), the number of categories $K$, and the parameters controlling the transition dynamics ($\beta, \sigma, \gamma, \rho$ for SEIRS). An alternative problem would primarily change the number of categories and also the parameters controlling the transition dynamics. For instance, the Malware Control example (Example 3 in the current manuscript) can be reasonably modelled as a population process with 2 categories (infected, not infected) and differing transition dynamics.
>
> So far, our experiments vary the population size $N$ and graph structure. We acknowledge our experiments can be made more extensive and we believe the interesting change to consider is varying the parameters controlling the transition dynamics.
>
> In summary, we commit to the following changes in an updated version of the manuscript:
>
> - We will state more clearly that the SEIRS problem is representative of most population processes
> - We will add additional experimental results covering different values for the transition parameters.
>
> As the experiments will take some time, we aim to have the updated manuscript prepared by the end of the rebuttal period.

---

> > ### Author Response · Authors · 2024-09-03
> > **Requested changes added**
> >
> > We have updated the manuscript with the requested changes. The writing of Section 5 now explains more clearly why the SEIRS problem is representative of most population processes. To make our experiments more extensive, we have also provided additional experimental results on two new transition parameter settings as well as added results on a new network.

---

> > > ### Comment · Reviewer_1uT5 · 2024-09-13
> > > **Response to authors**
> > >
> > > I thank the authors for the added details and the additional experiments. My concerns are addressed by the revision.

---

### Decision · Action_Editor_NFSj · 2024-11-04

**Recommendation:** Accept as is

**Comment:**

Two reviewers recommended an accept and one reviewer leaned towards a reject.

**Audience:**

Yes differential privacy and RL researchers will be interested in this submission.

**Claims And Evidence:**

Yes the claims made in the submission supported by accurate, convincing and clear evidence